# A curated dataset and lightweight deep learning framework for tea leaf disease classification

**Sakibul Hasan Chowdhury**☉¹⊚, **Md Shohel Arman**☉¹⊚*, **Masrafe Bin Hannan Siam**☉¹⊚, **Md Rayhan Khan**☉¹⊚, **Afia Hasan**¹‡, **Parvez Ahmed Moju Fahim**☉²‡

1 Data Science Lab, Department of Software Engineering, Daffodil International University, Daffodil Smart City, Birulia, Dhaka, Bangladesh, 2 Department of Computer Science and Engineering, Daffodil International University, Daffodil Smart City, Birulia, Dhaka, Bangladesh

⊚ These authors contributed equally to this work.
‡ AH and PAMF authors also contributed equally to this work.
* arman.swe@diu.edu.bd

## Abstract

Tea (*Camellia sinensis*) is the world's second most consumed beverage, enjoyed daily by more than two billion people. In Bangladesh, it serves as a cornerstone agricultural export and a major sector of the domestic economy. However, commercial tea cultivation remains highly vulnerable to fungal and pest-related diseases such as Blight, Red Rust, and Helopeltis which severely reduce crop yield and compromise leaf quality. While early detection is critical to preventing widespread outbreaks, traditional manual inspection is slow, subjective, and highly error-prone. Deep learning provides a scalable alternative, yet single-branch networks often struggle to capture both minute disease lesions and broader structural degradation simultaneously. To address this, we propose a Hybrid Feature Fusion architecture that runs two highly efficient feature extractors in parallel: EfficientNetV2-Small to isolate fine-grained local textures, and MobileNetV3-Small to capture the global structural context of the leaf. The models were trained and evaluated on a real-world dataset of 2,000 annotated images, evenly distributed across the four target classes (Blight, Red Rust, Helopeltis, and Healthy). Before training, the images underwent a standardized preprocessing pipeline including resizing to 224 × 224 pixels and normalization, supplemented by a dynamic augmentation strategy featuring random rotations, horizontal flips, and brightness adjustments to improve model robustness. The proposed hybrid framework achieved an outstanding peak classification accuracy of 96.80% alongside a macro Area Under the Curve (AUC) of 0.9980. To rigorously validate its performance, the hybrid model was benchmarked against six diverse architectures: a Vision Transformer (ViT-B16 at 76.40%), a Custom CNN (89.60%), MobileNetV3 (94.40%), ResNet50 (95.60%), DenseNet121 (96.40%), and EfficientNetV2-B3 (97.60%). Although EfficientNetV2-B3 achieved a marginally higher raw accuracy, the proposed dual-branch framework delivered a superior precision-recall balance and faster convergence stability. These findings demonstrate that the proposed hybrid

**Data availability statement:** The dataset comprising 2000 annotated tea leaf images was curated under real-world field conditions. It has been made available at https://data.mendeley.com/datasets/3x42rbj8yv/1. The computational code supporting the findings of this study is publicly accessible on GitHub: https://github.com/rayhankhan2192/Tea_Leaf_Disease_Model.

**Funding:** The author(s) received no specific funding for this work.

**Competing interests:** The authors have declared that no competing interests exist.

methodology is highly reliable and computationally balanced, making it an ideal candidate for integration into Internet of Things (IoT) edge devices for real-time disease monitoring in precision agriculture.

## Introduction

Tea (*Camellia sinensis*) is the world's second most popular beverage after water, cherished for its unique taste, aroma, and health benefits [1]. Every day, more than two billion people across the globe drink tea, making it one of the most significant agricultural products [2]. Rich in bioactive compounds such as polyphenols and antioxidants, tea is known to reduce cardiovascular risks, strengthen immunity, and promote overall well-being [3]. Beyond its nutritional value, tea also carries deep cultural meaning, often symbolizing hospitality, social connection, and tradition [4].

Bangladesh is among the leading tea producers in the world, with a history of cultivation dating back to the British colonial era [5]. At present, over 168 estates across Sylhet and Chittagong cover nearly 280,000 acres of land, contributing greatly to the national economy [6]. The tea sector also provides livelihoods for more than four million people, highlighting its role as a cornerstone of rural life [7]. Despite this importance, tea cultivation is constantly threatened by fungal, bacterial, and insect-related diseases [8]. Blight (Gray and Brown), Red Rust, and Helopeltis (tea mosquito bug) are among the most common issues, often leading to reduced yield, increased production costs, and compromised tea quality [9,10]. These diseases affect photosynthesis, weaken plant health, and ultimately degrade the flavor and aroma of processed tea [11,12]. In severe cases, unchecked outbreaks can result in widespread financial loss and supply chain disruptions [13].

Traditionally, tea disease detection has relied on manual field inspections. While widely practiced, this approach is labor-intensive, subjective, and often inaccurate [14]. The similarity of symptoms between different diseases makes diagnosis difficult, and the vast size of plantations makes continuous monitoring unrealistic [15]. These challenges emphasize the need for automated, intelligent, and scalable systems for early disease detection [16].

Recent advances in deep learning (DL) have shown encouraging results in crop disease detection for plants like rice, maize, and tomato [17]. However, the tea industry still lacks large, standardized, and annotated datasets. Most available datasets are either too small, poorly labeled, or collected under controlled conditions that do not reflect real-world challenges [18]. This scarcity limits the ability to train robust models and prevents meaningful benchmarking across studies [19].

Recent literature has extensively explored the deployment of lightweight convolutional neural networks, particularly MobileNetV3, for automated plant disease classification due to their low computational footprint. While foundational MobileNetV3 implementations have shown promise [20], recent advancements have primarily focused on optimizing this single-branch architecture by integrating attention mechanisms [21] or coupling it with 2D Discrete Wavelet Transforms (DWT) for

frequency-domain feature extraction [22,23]. Other studies have heavily relied on transfer learning to adapt MobileNetV3 to broader disease taxonomies [24]. However, despite these enhancements, relying on a single structural backbone inherently limits the network's ability to simultaneously capture minute, fine-grained localized lesions and broader global structural degradation. To overcome this critical bottleneck and advance beyond standard baseline implementations, our study introduces a novel Hybrid Feature Fusion architecture. Rather than optimizing a generic MobileNetV3 backbone in isolation, our methodology uniquely runs EfficientNetV2-Small in parallel to isolate highly detailed local textures, utilizing MobileNetV3-Small strictly as a global structural proxy. This dual-branch fusion directly addresses the representational limitations of existing standalone applications.

Recent advancements in 2024 and 2025 have predominantly focused on optimizing single-branch networks and exploring transformer-based architectures for tea leaf disease classification. For instance, recent studies have evaluated Vision Transformers (ViTs) alongside established Convolutional Neural Networks, demonstrating strong classification accuracy but noting that ViTs inherently require massive datasets to prevent overfitting [25]. Concurrently, other researchers have integrated advanced attention mechanisms into lightweight object detection frameworks like YOLOv8 and YOLO11 to improve lesion detection in complex backgrounds [26–29]. However, a persistent limitation across these highly optimized single-branch and transformer models is the difficulty in simultaneously decoupling minute, localized disease textures from the broader structural degradation of the leaf without significantly inflating computational demands. These constraints underscore the critical necessity of our proposed Hybrid Feature Fusion architecture. By running EfficientNetV2-Small and MobileNetV3-Small in parallel, our dual-branch framework effectively isolates fine-grained local symptoms while preserving global context, achieving robust diagnostic precision without the prohibitive data dependency or computational overhead of recent transformer alternatives.

In this study, we introduce a dataset comprising 2,000 images across four classes: Blight (Gray and Brown), Red Rust, Helopeltis, and Healthy leaves. The images were collected from tea gardens in Bangladesh under natural field conditions and carefully annotated by experts. To ensure model robustness, the images were standardized through resizing to 224×224 pixels and normalization, complemented by a dynamic data augmentation strategy including rotations, horizontal flips, and brightness adjustments to enrich dataset diversity [19].

To validate the usability of the dataset, a suite of deep learning models was evaluated, including our proposed Hybrid Feature Fusion Architecture, which achieved an overall accuracy of 96.80% in classifying tea leaf diseases.

## Distinctive contributions

The key contributions of this study are as follows:

- **Curated Dataset:** Creation of a high-quality, real-world dataset comprising 2,000 expertly annotated images spanning four distinct categories: Blight, Red Rust, Helopeltis, and Healthy tea leaves.

- **Robust Data Pipeline:** Implementation of a standardized preprocessing workflow coupled with dynamic, on-the-fly spatial and pixel-level augmentations—including rotations, horizontal flips, and brightness adjustments—to substantially enhance dataset diversity and model generalization [19].

- **Novel Architectural Framework:** Development and rigorous evaluation of a custom Hybrid Feature Fusion model that utilizes EfficientNetV2-S and MobileNetV3-S in parallel to process local textures and global structures, achieving a highly competitive peak classification accuracy of 96.80% and an outstanding macro AUC of 0.9980.

- **Comprehensive Benchmarking:** Establishment of a standardized comparative analysis evaluating seven diverse deep learning architectures—ranging from specialized CNNs and mobile-optimized networks to Vision Transformers—to provide a robust baseline for agricultural AI research.

- **Practical Relevance:** Demonstration of a highly efficient, dual-branch framework that optimally balances diagnostic precision with computational efficiency, highlighting its strong potential for integration into IoT and mobile edge platforms for real-time monitoring in commercial tea plantations [30].

## Integrated methodology

The complete workflow of the proposed tea leaf disease detection system is summarized in Fig 1. First, images of Blight, Red Rust, Helopeltis, and Healthy leaves are captured in tea gardens and stored in a centralized repository. After visual screening and expert annotation, the selected images are standardized to 224 × 224 pixels and pre-processed through RGB color space conversion and ImageNet-based normalization to ensure consistent input distribution across the network architectures.

The processed dataset is split into training (80%), validation (10%), and testing (10%) subsets using a stratified approach to maintain class proportions. To effectively increase variability and prevent model overfitting, only the training set is dynamically augmented utilizing the Albumentations framework. These specific augmentations include random rotations of up to 15 degrees, horizontal flips, and stochastic brightness and contrast adjustments. The validation and test sets remain strictly unaugmented, ensuring a fair and unbiased evaluation of the models' generalization capabilities.

The augmented training images are utilized to train and systematically evaluate an expanded suite of seven neural network architectures: a Custom CNN, ResNet50, DenseNet121, MobileNetV3, EfficientNetV2-B3, Vision Transformer (ViT-B16), and our proposed Hybrid Feature Fusion architecture. The networks are trained using the AdamW optimizer for enhanced weight decay management, coupled with a dynamically class-weighted Cross-Entropy loss function—alongside Focal Loss and Label Smoothing alternatives—to robustly address any underlying class imbalance. Following an early-stopping protocol driven by validation loss monitoring with a patience of 10 epochs, the models are rigorously evaluated on the unseen test set using accuracy, macro F1-score, Area Under the Curve (AUC), sensitivity, and specificity metrics.

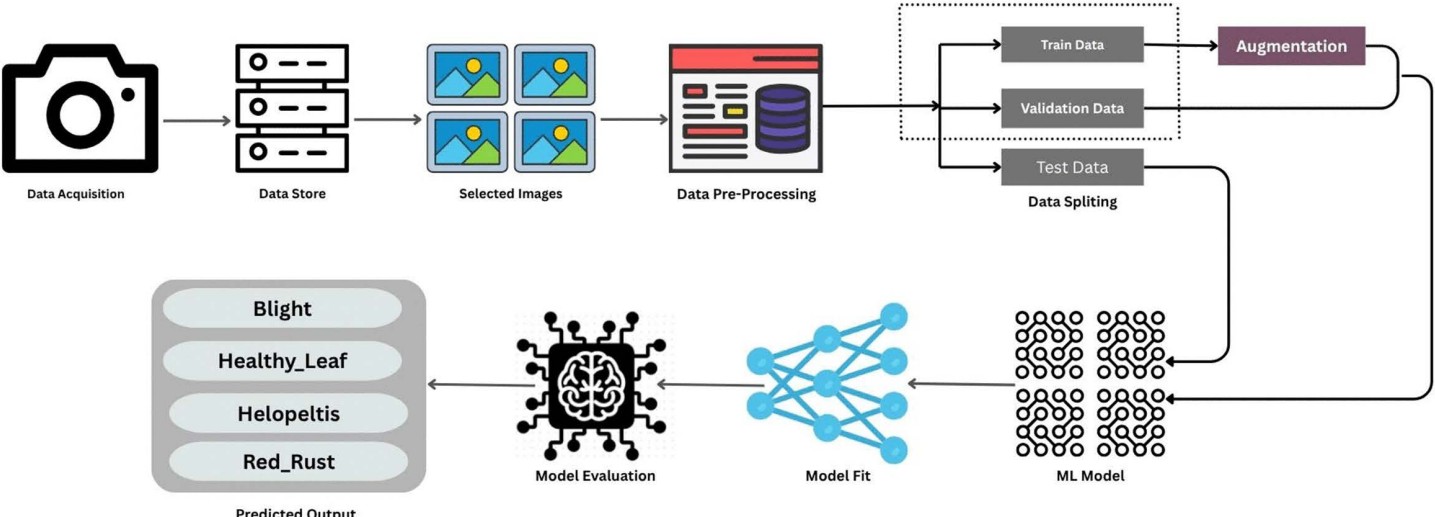

**Fig 1. Overview of the integrated methodology for tea leaf disease classification, from data acquisition and preprocessing to model training, evaluation, and final prediction.**

## Data collection

The images used in this study were collected from major tea-growing areas of Bangladesh. These images were manually captured between April 7, 2025, and June 23, 2025, from the tea gardens in the Sylhet region, specifically within the Habiganj district. These regions were selected as they represent some of the oldest and largest tea plantations in the country and routinely experience prevalent diseases such as Blight, Red Rust, and Helopeltis.

Field visits were carried out during regular working hours, and leaves were photographed directly on the plants under natural lighting conditions. To ensure the dataset accurately reflects the complexities of real-world agricultural environments, images were captured across a diverse range of natural conditions, including bright midday sunlight, overcast skies, and partial canopy shading. These variations introduced complex shadows and specular highlights, while the presence of background clutter—such as overlapping leaves, branches, and soil—requires the models to successfully isolate pathological symptoms from noisy environments. The dataset includes leaves photographed from multiple orientations and distances, capturing diseases at various stages of severity, from early-stage necrotic spots to advanced structural degradation. After removing blurred or ambiguous samples, a final set of 2,000 images was retained, spanning four target classes: Blight (Gray and Brown), Red Rust, Helopeltis, and Healthy. All images in this dataset were captured using an iPhone 12 Pro Max, equipped with a 12-megapixel triple-camera system (comprising Ultra-Wide, Wide, and Telephoto lenses). The camera produced high-resolution images ranging from 1080x1080 to 3024x3024 pixels, ensuring the detailed capture of microscopic leaf textures, subtle disease symptoms, and complex background conditions. The device's advanced image processing algorithms enabled accurate color reproduction, sharpness, and clarity under natural field conditions.

To ensure high fidelity, a rigorous multi-stage annotation protocol was implemented. All 2,000 images were independently inspected by agricultural domain experts specializing in plant pathology. The classification criteria were strictly based on established phenotypic markers: characteristic necrotic lesion patterns for Blight, orange-red algal spots for Red Rust, and distinct angular feeding punctures for Helopeltis. In instances of disagreement, a third agriculturalist reviewed the images to reach a final consensus. This systematic cross-validation ensures that the ground truth labels entered the training pipeline without subjective bias.

To contextualize the value of our newly curated dataset, it is imperative to compare it against existing state-of-the-art (SOTA) tea leaf disease datasets such as teaLeafBD and TDPD. The primary strength of our dataset lies in its authentic capture of high-variance natural conditions and its relative class equilibrium (Table 1). While larger public datasets often suffer from inconsistent annotation or laboratory-controlled lighting, our dataset prioritizes environmental realism and diagnostic accuracy. To further address the slight numerical variations between classes, our training pipeline utilizes automated class-weight computation to ensure the models maintain unbiased gradients across all categories.

**Table 1. Detailed summary of the curated tea leaf dataset attributes.**

| Class Name | Total Images | Source Resolution | Format | Color Space |
|---|---|---|---|---|
| Blight | 510 | 1024 × 1024 | JPG | RGB |
| Red Rust | 475 | 1024 × 1024 | JPG | RGB |
| Helopeltis | 490 | 1024 × 1024 | JPG | RGB |
| Healthy | 525 | 1024 × 1024 | JPG | RGB |
| **Total** | **2000** | | | |

## Image pre-processing and data augmentation

A curated dataset comprising 2,000 tea leaf images across four distinct classes (Blight, Red Rust, Helopeltis, and Healthy) was collected under natural field conditions in the tea gardens of Bangladesh. To ensure architectural compatibility and stable gradient convergence, all raw images were integrated into a custom PyTorch data loading pipeline where they were resized to a uniform dimension of 224 × 224 pixels. During this stage, images were converted to the RGB color space and standardized using ImageNet normalization metrics (mean: 0.485, 0.456, 0.406; standard deviation: 0.229, 0.224, 0.225) to ensure an optimal input distribution for the deep learning models. To actively combat class imbalance, increase dataset diversity, and prevent model overfitting, we implemented a dynamic, on-the-fly data augmentation strategy using the Albumentations library. This strategy was applied exclusively to the training phase to help the models generalize across variable outdoor illumination and leaf orientations. Specifically, the pipeline utilized a stochastic sequence of spatial and pixel-level transformations: random rotations of up to 15 degrees were applied with a 70% probability, horizontal flips with a 50% probability, and random brightness and contrast adjustments with a 20% probability (Fig 2). This approach ensures that the networks encounter a high-variance representation of disease symptoms—such as necrotic spots and fungal rust—during every training epoch. In contrast, the validation and test sets remained strictly unaugmented, utilizing only resizing and normalization to provide an unbiased assessment of the models' performance on real-world, unseen data.

## Hyperparameter configuration and training protocol

To ensure a rigorous and mathematically fair comparative analysis, a standardized hyperparameter protocol was enforced across all evaluated architectures. Every model—including the baseline CNNs, the Vision Transformer, and our proposed Hybrid Feature Fusion network—was trained under identical computational constraints. We utilized the AdamW optimizer with a uniform initial learning rate of 0.001. The training process was configured with a fixed batch size of 16, utilizing a Cross-Entropy loss function dynamically scaled with balanced class weights to account for class-level disparities. To mitigate overfitting and ensure optimal generalization, an Early Stopping callback was universally implemented with a patience of 10 epochs. Consequently, while individual architectures naturally converged and halted at different total epoch counts—ranging from 17 to 30 epochs—the foundational training constraints and regularization strategies remained identical, guaranteeing a rigorous benchmarking environment.

All computational experiments were executed on a dedicated workstation equipped with an NVIDIA GeForce RTX 3060 GPU (6 GB VRAM) and an AMD Ryzen 7 5700X CPU, supported by 16 GB of RAM. The runtime environment utilized GPU acceleration via CUDA to optimize training efficiency. The total training time for the proposed Hybrid Feature Fusion model to reach convergence was approximately one hour, reflecting the architectural efficiency of the dual-branch framework. This final configuration was selected based on empirical baseline testing where an initial learning rate of 0.001 and a batch size of 16 consistently yielded the lowest validation loss and highest macro AUC while maintaining gradient stability during the early training phases.

## Proposed hybrid feature fusion architecture and implementation

To effectively capture both the fine-grained symptoms of localized lesions and the broader structural degradation of the tea leaves, we designed a novel hybrid feature fusion architecture. Unlike traditional single-branch networks, this proposed model employs a dual-branch feature extraction strategy running in parallel (Figure 3). The first branch utilizes the pre-trained EfficientNetV2-S backbone [31], which leverages Fused-MBConv blocks to efficiently extract highly detailed local textures, such as minute fungal spots and rust patches. Simultaneously, the second branch employs the pre-trained MobileNetV3-Small architecture [32] to serve as a feature proxy, capturing the global context and overall leaf morphology.

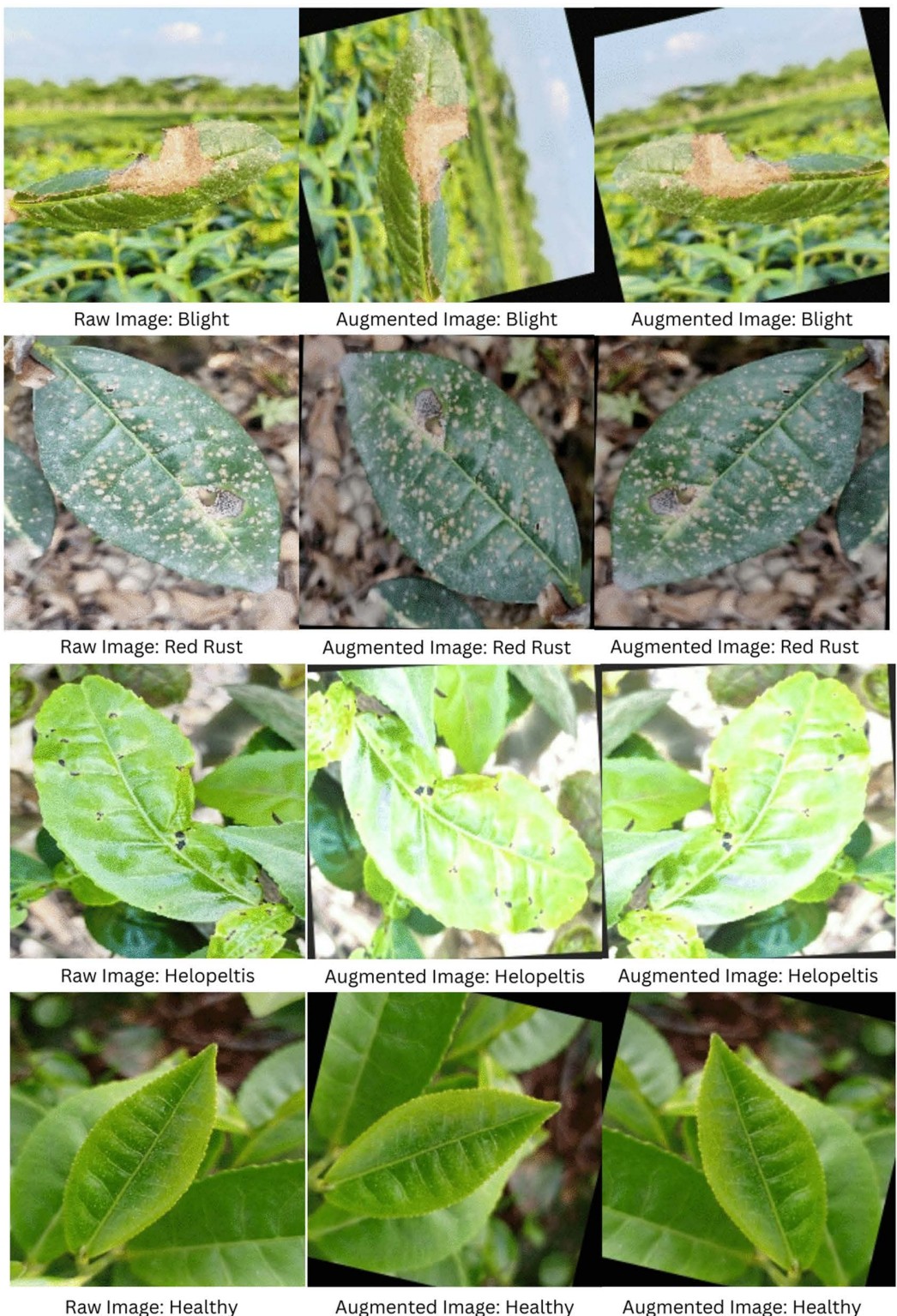

**Fig 2. Visual progression of the data pipeline.** Representative samples of Blight, Red Rust, Helopeltis, and Healthy classes showing raw field captures (left) alongside subsequent iterations of the dynamic augmentation pipeline (middle and right), which utilize stochastic rotations, horizontal flips, and brightness/contrast adjustments to enhance model generalization.

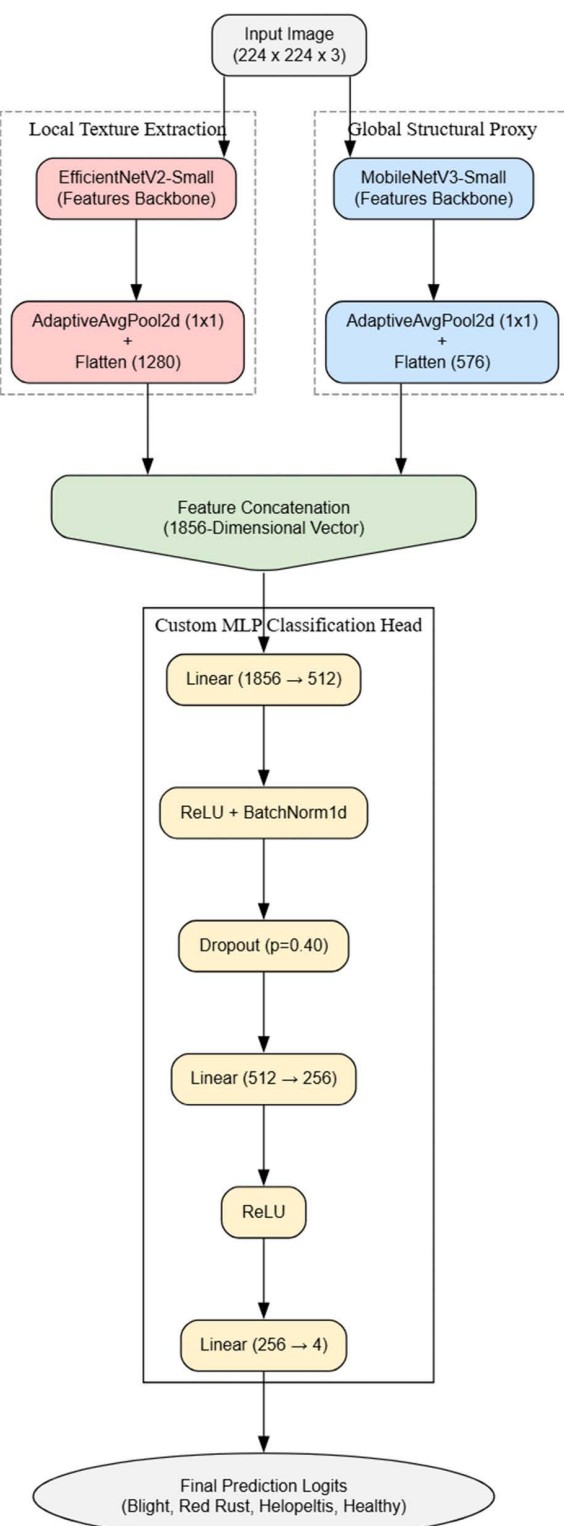

**Fig 3. Flowchart of the Proposed Hybrid Feature Fusion Architecture and Implementation.** The diagram illustrates the dual-branch feature extraction strategy running in parallel.

Outputs from both pathways are processed through an adaptive average pooling layer to generate dense, one-dimensional feature representations: a 1280-dimensional vector from the EfficientNetV2-S branch and a 576-dimensional vector from the MobileNetV3-Small branch. These representations are then fused via concatenation into a comprehensive 1856-dimensional feature space.

To map these combined features to the target disease classes, we engineered a custom sequential multi-layer percep-tron (MLP) classification head. The fused vector is initially projected into a 512-node linear layer, followed by an in-place Rectified Linear Unit (ReLU) activation and 1D batch normalization to stabilize deep feature transitions. A 40% dropout rate is subsequently applied to mitigate overfitting. The data is then compressed into a 256-node linear layer, followed by another ReLU activation, before a terminal linear projection yields the raw prediction logits for the four disease categories. The fundamental mathematical operations underlying this hybrid forward pass—specifically feature concatenation, the fully connected linear transformation, and batch normalization—are defined as follows:

**Feature Concatenation:**

$$f_{fused} = [f_{cnn} \parallel f_{mob}]$$

where $f_{cnn}$ and $f_{mob}$ represent the flattened feature vectors from the EfficientNetV2-S and MobileNetV3-Small branches, respectively.

**Linear Transformation (Fully Connected Layer):**

$$z = Wx + b$$

where $W$ denotes the learned weight matrix, $x$ is the input feature vector, and $b$ represents the bias term.

**Batch Normalization:**

$$y_i = \gamma \left( \frac{x_i - \mu_B}{\sqrt{\sigma_B^2 + \epsilon}} \right) + \beta$$

where $\mu_B$ and $\sigma_B^2$ are the mini-batch mean and variance, $\gamma$ and $\beta$ are learnable scale and shift parameters, and $\epsilon$ is a small constant ensuring numerical stability.

## Compared models

To validate the effectiveness of the proposed Hybrid Feature Fusion Architecture, its performance was compared against several state-of-the-art deep learning architectures. These models were selected due to their proven performance in image classification and their ability to generalize across complex agricultural datasets. The compared architectures include:

• **Custom CNN Model**

• **ResNet50**

• **DenseNet121**

• **MobileNetV3**

• **EfficientNetV2-B3**

• **Vision Transformer (ViT-B16)**

Transfer learning was employed across all pre-trained architectures, where the original classification heads were replaced with custom multi-layer perceptrons (MLPs) specifically fine-tuned for the four target tea leaf disease categories.

## Custom CNN Model Architecture and Implementation

To establish a domain-specific baseline for tea leaf disease classification, we designed a custom Convolutional Neural Network (CNN) from scratch. The feature extraction backbone is constructed from six sequential convolutional blocks [33,34]. The network initially applies a 3 × 3 convolution to the input images to extract 32 feature channels, which is then expanded to 64 channels for the subsequent five convolutional layers. A zero-padding of 1 is applied across all convolutions to preserve spatial dimensions prior to downsampling. To introduce non-linearity and reduce spatial resolution, each convolutional layer is followed by an in-place Rectified Linear Unit (ReLU) activation and a 2 × 2 max pooling operation.

To promote stable gradient flow and optimal convergence, all convolutional weights are initialized using the Kaiming normal distribution. Following the final max pooling step, an adaptive average pooling layer standardizes the output to a 1 × 1 spatial resolution. This flattened 64-dimensional feature vector is then passed into a streamlined classification head, comprising a fully connected layer with 64 units, a ReLU activation, and a final linear layer that outputs the raw logits for the target disease classes. The fundamental mathematical operations governing this network's forward pass—specifically convolution, ReLU activation, max pooling, and the final softmax probability distribution—are defined as follows:

**Convolution:**

$$y_{i,j,k} = \sum_{m,n,c} w_{m,n,c,k}\, x_{i+m,\, j+n,\, c} + b_k$$

**ReLU:**

$$\phi(u) = \max(0, u)$$

**Max Pooling:**

$$y_{i,j,k} = \max_{(m,n) \in \mathcal{R}(i,j)} x_{m,n,k}$$

**Softmax (output):**

$$p_c = \frac{e^{z_c}}{\sum_{r=1}^{C} e^{z_r}}$$

## ResNet50 model architecture and implementation

The ResNet50 architecture was evaluated to leverage its deep, 50-layer residual framework for tea leaf disease classification [35]. A defining characteristic of this network is its use of bottleneck blocks—a sequence of 1 × 1, 3 × 3, and 1 × 1 convolutions designed to efficiently capture complex spatial hierarchies. Crucially, ResNet50 utilizes skip connections to bypass certain layers, a structural innovation that effectively mitigates the vanishing gradient problem in deep networks. Instead of learning a direct unreferenced mapping $H(x)$, the architecture learns a residual mapping defined mathematically as:

$$H(x) = F(x) + x$$

where $F(x)$ represents the residual function. Within the network's working procedure, each block computes its output as:

$$y = F(x, W) + x$$

where $F(x, W)$ denotes the convolutional transformations governed by weights $W$. This design allows gradients to flow unimpeded through the skip connections during backpropagation.

Following the core convolutional stages and global average pooling [36], we modified the network's classification mechanism to better suit our specific dataset. We replaced the default fully connected layer with a custom Multi-Layer Perceptron (MLP) head. This custom head first projects the deep features into a 1024-dimensional space, applying an in-place ReLU activation and 1D batch normalization to stabilize feature transitions. To prevent overfitting, a 40% dropout rate is applied before mapping the data to a 512-dimensional layer. This is followed by another ReLU activation and a second batch normalization step intended to prevent gradient collapse. Finally, a secondary dropout of 20% is applied before the terminal linear layer outputs the predictive logits for the target disease classes.

### DenseNet121 model architecture and implementation

The DenseNet121 architecture was integrated into our framework to exploit its highly efficient feature reuse capabilities [37]. Comprising 121 layers, this network distinguishes itself through dense connectivity, where each layer receives direct inputs from all preceding layers within its specific block. This foundational mechanism is mathematically formalized as:

$$x_l = H_l([x_0, x_1, \ldots, x_{l-1}])$$

where the brackets denote the concatenation of previous feature maps. The network expands based on a predefined growth rate that dictates the number of new feature maps added per layer. Because features are continuously reused throughout the network, the architecture reduces the number of required parameters while simultaneously enhancing gradient flow.

In practice, the input data progresses through dense blocks composed of convolutional layers coupled with batch normalization and ReLU activations. Between these dense blocks, transition layers featuring a 1 × 1 convolution and 2 × 2 average pooling are applied to sequentially compress the feature dimensions. For our specific classification task, we fine-tuned a pre-trained DenseNet121 backbone and customized its terminal classification stage. The native classification head was replaced with a tailored sequential block that first projects the deep features into a 512-dimensional linear layer. To accelerate convergence and stabilize the deep feature representations, this layer is immediately followed by an in-place ReLU activation and 1D batch normalization. Finally, to actively mitigate overfitting during training, a 50% dropout layer is applied before the ultimate linear layer maps the representations directly to the four target tea leaf disease classes.

### MobileNetV3 model architecture and implementation

The MobileNetV3 architecture was evaluated in our study to take advantage of its balance between classification accuracy and computational efficiency [32,38,39]. This network integrates depthwise separable convolutions with inverted residual frameworks, while refining its layer structures through Hardware-Aware Neural Architecture Search (NAS). The network draws its core efficiency from depthwise separable convolutions, which reduce computational overhead compared to standard convolutions. The operation for depthwise convolution is mathematically expressed as:

$$y_{i,j,k} = \sum_{m,n} K_{m,n,k}\, x_{i+m,j+n,k}$$

To facilitate complex feature learning without destroying gradients, it employs inverted residuals with linear bottlenecks, defined as:

$$y = W_2 \cdot DW(W_1 \cdot x)$$

Furthermore, the architecture embeds Squeeze-and-Excitation (SE) blocks to apply channel-wise attention, highlighting the most informative feature maps through the computation:

$$s_c = \sigma\big(W_2\,\delta(W_1 z_c)\big), \quad z_c = \frac{1}{H \times W} \sum_{i,j} x_{i,j,c}$$

Our methodology specifically employs the pre-trained MobileNetV3-Large backbone to ensure the capture of deep visual features from tea leaf lesions. To tailor this backbone for our diagnostic task, we discarded the default classifier and implemented a custom sequential processing head. The 960-dimensional feature maps extracted by the backbone are first condensed using a 1 × 1 adaptive average pooling layer and flattened. The data then flows into a 512-node linear layer, followed by an in-place Hardswish activation function and 1D batch normalization to stabilize feature distributions and accelerate convergence. To prevent overfitting, we enforce a 30% dropout rate before passing the data into a subsequent 256-node linear layer. After a final Hardswish activation step, a terminal linear layer maps the extracted patterns to the four target disease classes.

### EfficientNetV2-B3 model architecture and implementation

To maximize classification accuracy while optimizing computational resources, we integrated the EfficientNetV2-B3 architecture into our study [31]. The EfficientNet family is distinguished by its compound scaling method, which uniformly scales network depth, width, and image resolution [40]. This optimal scaling is mathematically formulated using a compound coefficient $\phi$ to determine the network's dimensions:

$$d = \alpha^{\phi}, \quad w = \beta^{\phi}, \quad r = \gamma^{\phi}$$

subject to the constraint $\alpha \cdot \beta^2 \cdot \gamma^2 \approx 2$ (where $\alpha \geq 1, \beta \geq 1, \gamma \geq 1$), where $\alpha$, $\beta$, and $\gamma$ dictate resource allocation across depth ($d$), width ($w$), and resolution ($r$).

The V2 iteration enhances training speed and parameter efficiency by strategically replacing standard depthwise convolutions with Fused-MBConv blocks in its early layers [31]. Our implementation incorporates a robust fallback mechanism that attempts to load the EfficientNetV2-B3 backbone but seamlessly reverts to the standard EfficientNet-B3 architecture to ensure continuous execution across different environment setups. To adapt this backbone for the nuances of our tea leaf dataset, we replaced the native classifier with a custom sequential multi-layer perceptron (MLP) head. The deep features extracted by the backbone are first projected into a 1024-dimensional linear layer, paired with an in-place ReLU activation and 1D batch normalization to stabilize gradient flow. To mitigate overfitting, we apply a 30% dropout rate at this stage. The data is then compressed into a 512-node linear layer, subjected to another round of ReLU activation and batch normalization, and regularized with a 15% dropout. Finally, a terminal linear projection maps the refined feature set to the four target tea leaf disease classes.

## Vision transformer (ViT-B16) architecture and implementation

To broaden our architectural evaluation and explore non-convolutional paradigms, we incorporated the Vision Transformer (ViT-B16) into our study [41]. Unlike standard Convolutional Neural Networks that rely on localized receptive fields to build feature maps, ViT fundamentally treats an image as a sequence of flattened 2D patches—in this case, 16 × 16 pixels each. These patches are linearly projected into a 1D sequence of token embeddings and supplemented with positional embeddings to retain spatial relationships. The architecture heavily relies on the Multi-Head Self-Attention (MHSA) mechanism to capture global dependencies and contextual relationships across the entire image simultaneously. The core self-attention operation is mathematically defined as:

$$\text{Attention}(Q, K, V) = \text{softmax}\left(\frac{QK^T}{\sqrt{d_k}}\right) V$$

where $Q$, $K$, and $V$ represent the query, key, and value matrices, respectively, and $d_k$ acts as a scaling factor to stabilize gradients during training.

In our implementation, we utilized the pre-trained ViT-B/16 backbone to harness its powerful global representations. However, to specialize the model for the detection of our specific tea leaf diseases, we discarded the default classification head and integrated a custom multi-layer perceptron (MLP). The deep features extracted by the transformer are first passed through a 512-node linear layer. Instead of a standard ReLU, this layer employs a Gaussian Error Linear Unit (GELU) activation function, which provides smoother non-linearities and is highly synergistic with transformer-based architectures. To mitigate the risk of overfitting—a common issue when applying data-hungry transformers to smaller datasets—a 20% dropout rate is enforced before the data passes through the final linear projection. This terminal layer maps the generalised token representations directly to the four distinct disease categories.

 

## Experimental result

### Training & Validation setup

As illustrated in Table 2, to ensure robust and unbiased model evaluation, the curated dataset was partitioned into 80% training, 10% validation, and 10% testing subsets using a stratified split to preserve class distribution across all experimental phases (Fig 4). All computational experiments were executed within the PyTorch deep learning framework. To optimize the network parameters and improve generalization, we utilized the AdamW optimizer with an initial learning rate of 0.001, capitalizing on its superior weight decay handling compared to standard Adam optimization. The architectures were trained utilizing a batch size of 16 over a maximum configuration of 20 epochs. Furthermore, to address class imbalances within the dataset, the training pipeline dynamically computes balanced class weights and integrates them directly into the cross-entropy loss function, with built-in mechanisms to utilize Focal Loss or Label Smoothing as required. Finally, to prevent the networks from overfitting, a strict early stopping protocol was implemented; the training cycle is automatically terminated if the validation loss fails to improve over a patience window of 10 consecutive epochs.

**Table 2. Stratified dataset split (80-10-10) preserving class equilibrium.**

| Class Name | Training (80%) | Validation (10%) | Testing (10%) |
|---|---|---|---|
| Blight | 408 | 51 | 51 |
| Red Rust | 380 | 47 | 48 |
| Helopeltis | 392 | 49 | 49 |
| Healthy | 420 | 53 | 52 |
| **Total** | **1600** | **200** | **200** |

Dataset Split Distribution (Donut Chart)

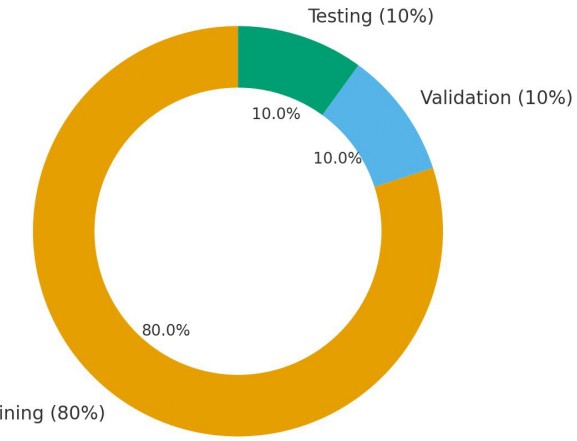

**Fig 4. Dataset split distribution showing the proportion of images used for training (80%), validation (10%), and testing (10%).**

## Models evaluation results

### Custom CNN model

To evaluate how well the Custom Convolutional Neural Network (CNN) architecture performs, we ran a series of experiments using our prepared dataset. We carefully applied optimization and regularization techniques to keep the training process stable and to stop the model from simply memorizing the training images. Throughout the training cycles, the model displayed a solid and consistent learning pattern. The training accuracy climbed steadily, eventually reaching 91.80%, while the training loss dropped significantly to about 0.24. At the same time, the validation metrics showed that the model was generalizing quite well to unseen data. The validation accuracy peaked at a strong 89.60% during the later epochs before settling at 85.20% by the end of training. As illustrated in Fig 5 and Fig 6, while a slight gap naturally emerges between the training and validation curves toward the end, the validation loss remains largely under control, finishing near 0.55. It is also worth highlighting that the model achieved an excellent Area Under the Curve (AUC) score, peaking at 0.976 on the validation set. These updated metrics clearly confirm that our custom CNN is highly effective at picking out the vital spatial features from tea leaves, proving it to be a dependable tool for disease classification tasks.

### ResNet50

We evaluated the pre-trained ResNet50 architecture using transfer learning to determine how well its deep residual framework could handle our tea leaf disease dataset. Unlike early baseline runs where the model had difficulty generalizing, our refined training process yielded fantastic results, proving that the network can indeed adapt to the domain-specific features of tea leaves when properly optimized. Looking at the updated metrics in Fig 7 and Fig 8, the learning curves highlight a highly effective training phase. Training accuracy climbed steadily to reach just over 95%, while the training loss dropped smoothly from 0.67 down to 0.16. The validation metrics closely mirrored this success. Validation accuracy consistently tracked in the 90s, ultimately peaking at an impressive 95.60%. While there was a bit of expected volatility in the validation loss—such as a brief spike around epoch 19—it generally remained low, ultimately settling near 0.17. Furthermore, the model's discriminative power was stellar; the validation Area Under the Curve (AUC) consistently stayed

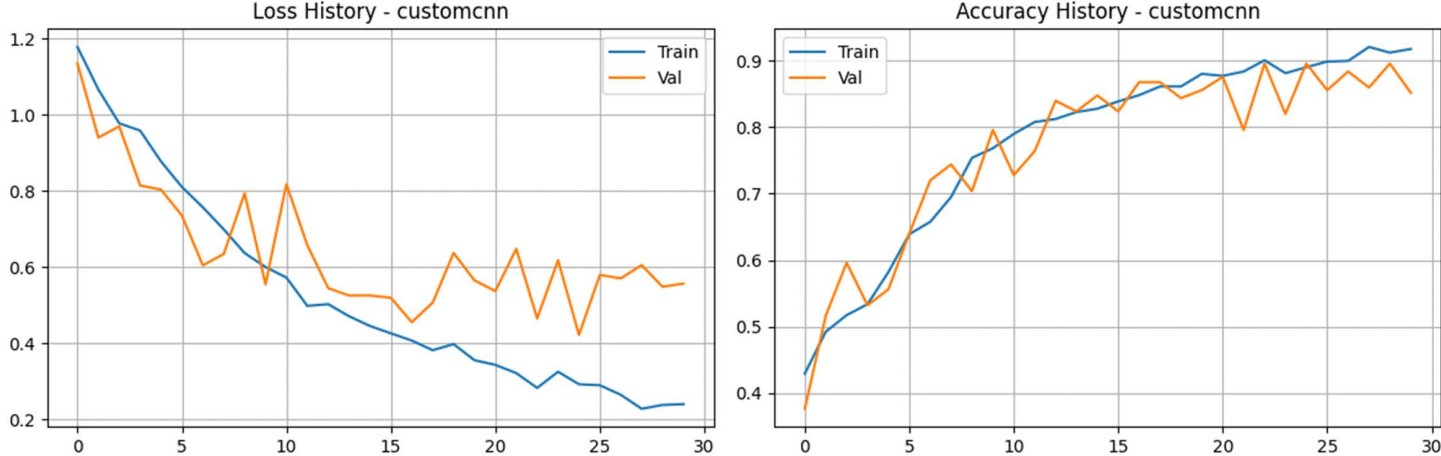

**Fig 5. Training and validation performance of the customized CNN.** The plots illustrate the loss history (left) and accuracy history (right) over 30 epochs, demonstrating a steady convergence in training while revealing characteristic fluctuations in validation metrics typical of learning on high-variance natural field imagery.

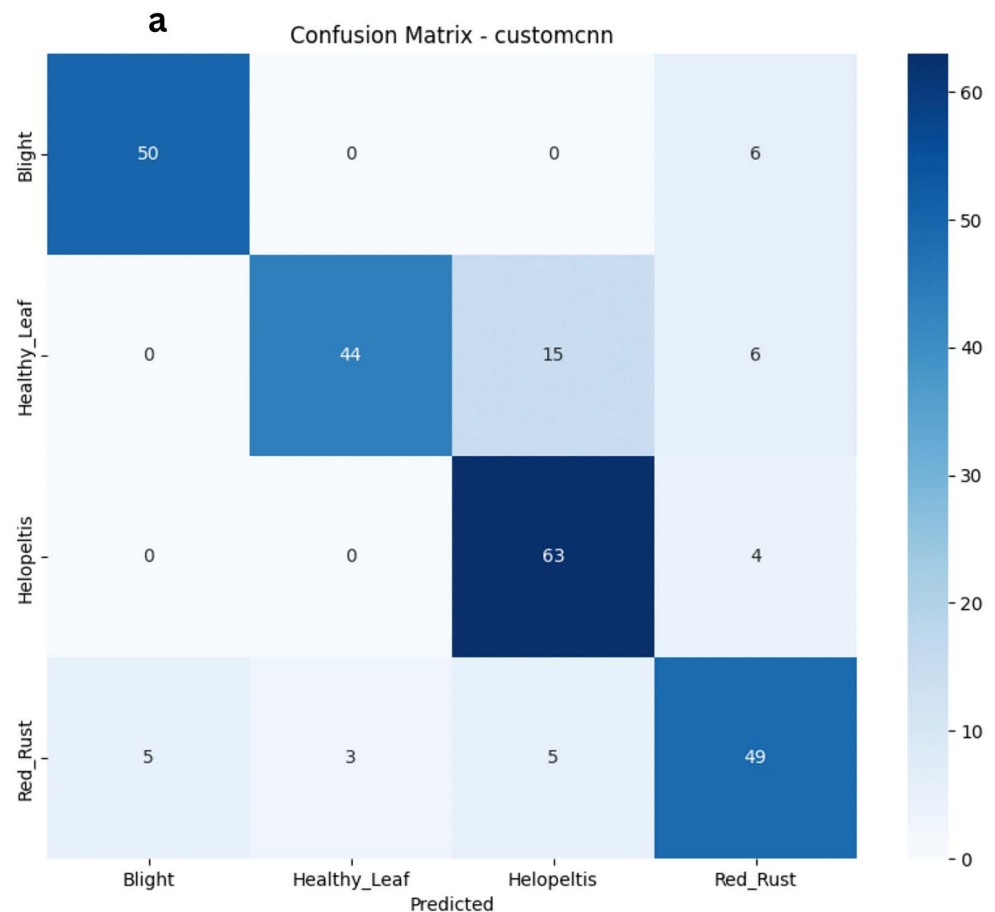

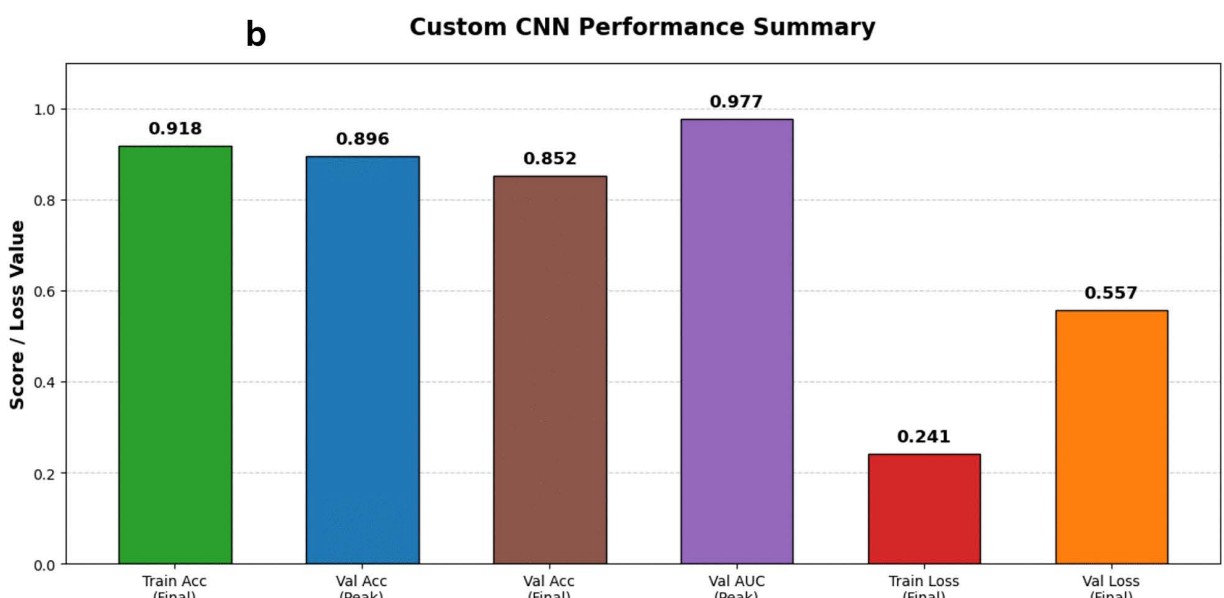

**Fig 6. Performance analysis of the Custom CNN baseline architecture.** a) Confusion matrix for the Custom CNN Model on the stratified test set. b) Bar Chart for the Custom CNN Model.

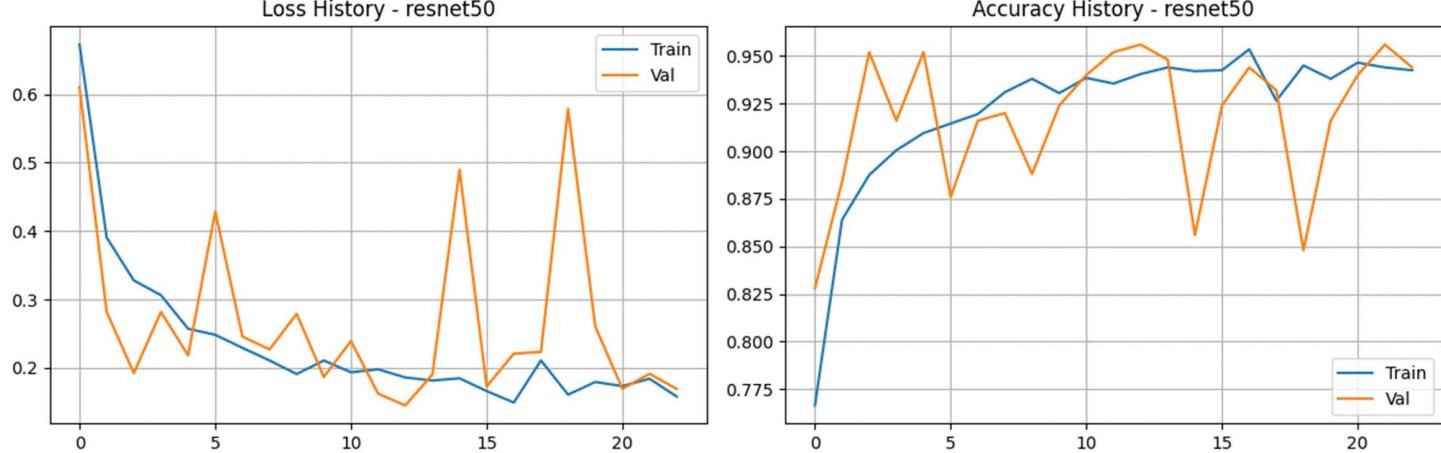

**Fig 7. Training and validation performance of the refined ResNet50.** The plots show the loss history (left) and accuracy history (right) for the ResNet50 model with its custom MLP head.

above 0.98, reaching as high as 0.997. Ultimately, ResNet50 demonstrated robust learning behavior and excellent classification capabilities. While its deeper, parameter-heavy structure naturally demands higher computational costs and slightly longer training times compared to lightweight models like MobileNetV3, its ability to successfully capture complex spatial patterns makes it a highly effective and accurate tool for this diagnostic task.

### DenseNet121

To explore the benefits of feature reuse, we fine-tuned the DenseNet121 architecture on our tea leaf disease dataset. The defining characteristic of this network is its dense connectivity: each layer directly receives feature maps from all preceding layers. This setup is highly effective at keeping gradients flowing smoothly throughout the network while minimizing redundant feature learning. The updated training metrics highlight a highly successful learning process. As shown in Fig 9 and Fig 10, both training and validation accuracies climbed rapidly in the early stages and stabilized at an impressive 94% to 96% range. Validation accuracy ultimately peaked at 96.40%, while the validation loss dropped down to an excellent 0.12. The close tracking between the training and validation curves indicates that the model achieved robust generalization without suffering from significant overfitting. Furthermore, the model's discriminative ability was exceptional, with the validation Area Under the Curve (AUC) consistently staying above 0.99 in the latter half of training, peaking at nearly 0.998. Ultimately, DenseNet121 proved to be an incredibly strong performer. It delivered highly competitive accuracy and stable learning dynamics, proving its reliability for this classification task, even though it naturally demands a bit more computational overhead compared to strictly lightweight models like MobileNetV3.

### MobileNetV3

The fine-tuned MobileNetV3 architecture demonstrated strong and reliable performance on our tea leaf disease dataset. By taking advantage of depthwise separable convolutions and squeeze-and-excitation blocks within its lightweight framework, the model successfully secured high accuracy rates without demanding excessive computational power. Looking at the updated training dynamics in Fig 11 and Fig 12, the network learned rapidly. Training accuracy climbed steadily to peak at 97.25%, while the training loss dropped as low as 0.08. On the validation side, we did observe a sharp spike in loss during the very first epoch a common occurrence as weights make their initial large adjustments. However, it swiftly

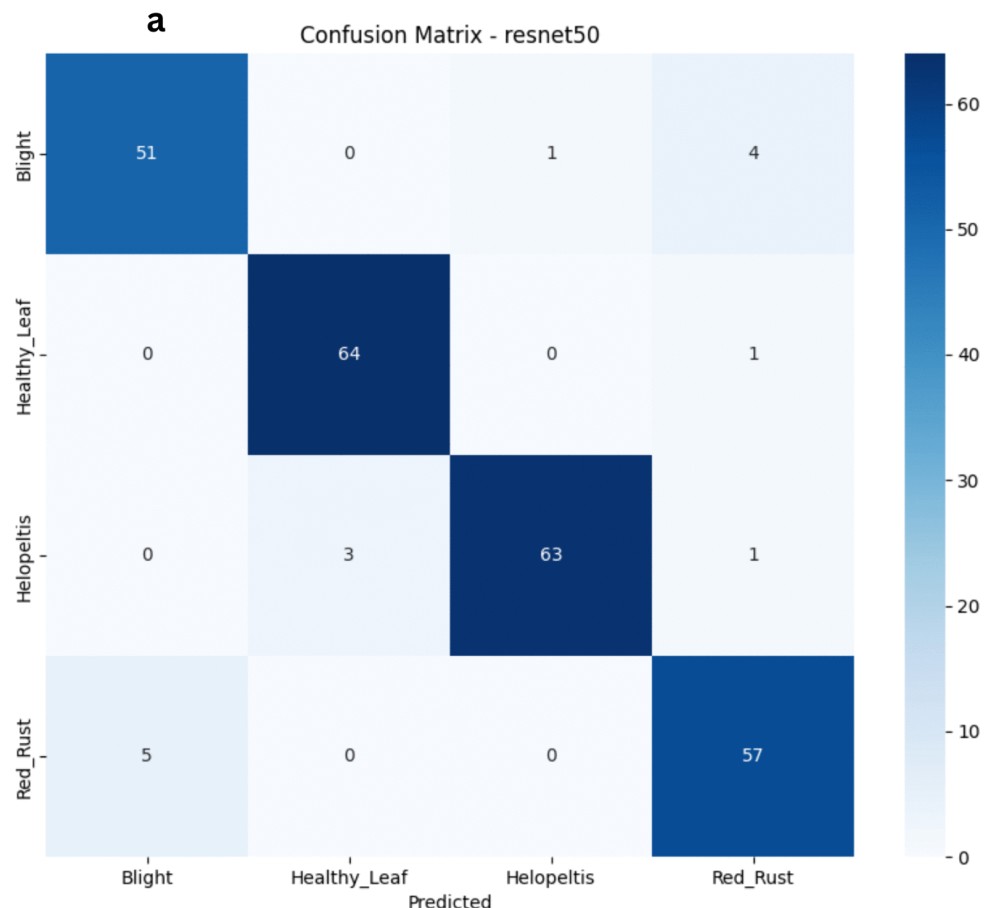

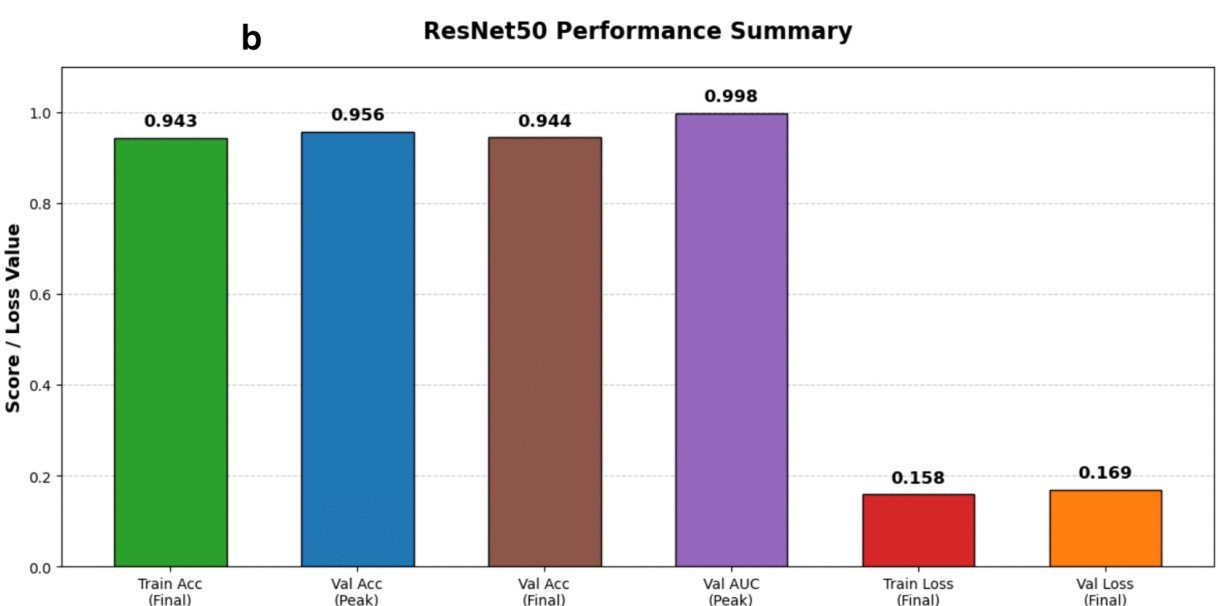

**Fig 8. Performance analysis of the ResNet50 baseline architecture.** a) Confusion matrix for the ResNet50 Model on the stratified test set. b) Bar Chart for the ResNet50 Model.

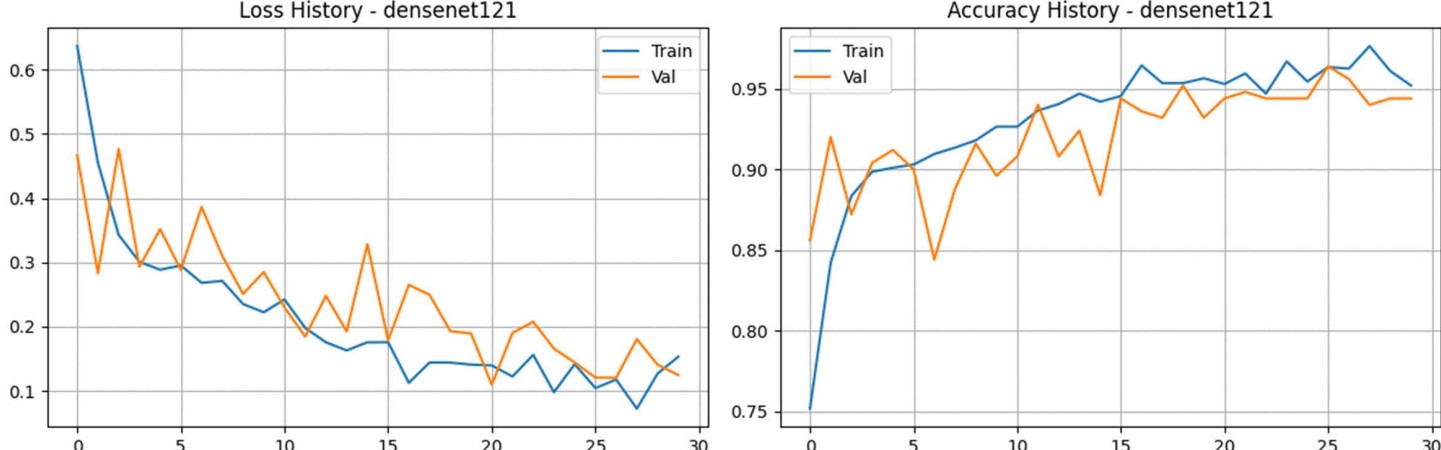

**Fig 9. Training and validation performance of the DenseNet121 model.** The plots show the loss history (left) and accuracy history (right) over 30 epochs, demonstrating high discriminative power with a final validation accuracy.

corrected itself, dropping down and settling into a very stable range near 0.18 to 0.20. Validation accuracy quickly stabilized in the 92% to 94% range, eventually hitting a peak of 94.40%. It is also important to highlight the model's outstanding discriminative power; the validation Area Under the Curve (AUC) consistently stayed above 0.98, reaching an impressive high of 0.995. Ultimately, MobileNetV3 proved to be highly efficient. Its smooth convergence, exceptionally high AUC, and incredibly low computational footprint reaffirm that it is a practical, scalable, and highly capable choice for real-time agricultural disease detection in the field.

### EfficientNetV2-B3

To expand our comparative analysis during the revision process, we incorporated the EfficientNetB3 architecture into our experiments. This specific model is highly regarded for its compound scaling method, which uniformly scales network depth, width, and image resolution to maximize performance without causing an explosion in computational overhead. The training logs reveal a highly efficient and remarkably rapid learning phase that converged in just 17 epochs (Fig 13 and Fig 14). The network grasped the training data almost immediately; training accuracy surged to a peak of 98.35%, while the training loss plummeted to a low of roughly 0.06. More importantly, the validation metrics proved that the model was genuinely generalizing rather than just memorizing the data. Validation accuracy started strong at 94% right out of the gate and ultimately hit an outstanding peak of 97.60%. The validation loss remained tightly controlled throughout the run, dropping as low as 0.11 before finishing near 0.14. The model's ability to discriminate between the four tea leaf categories was exceptional. Across the entire training cycle, the validation Area Under the Curve (AUC) never once dropped below 0.99, hitting a near-perfect high of 0.9968. These results clearly demonstrate EfficientNetB3's phenomenal diagnostic capabilities. It strikes a brilliant balance, offering accuracy that rivals the absolute best-performing models in our study while remaining highly efficient.

### Vision transformer (ViT-B16)

To broaden the scope of our architectural comparison, we also evaluated the Vision Transformer (ViT-B16) model. Unlike traditional convolutional networks that rely on local receptive fields, ViT treats image patches as a sequence of tokens and applies self-attention mechanisms to capture global dependencies. However, the training dynamics revealed that this architecture faced challenges adapting to our specific dataset. The learning process was notably

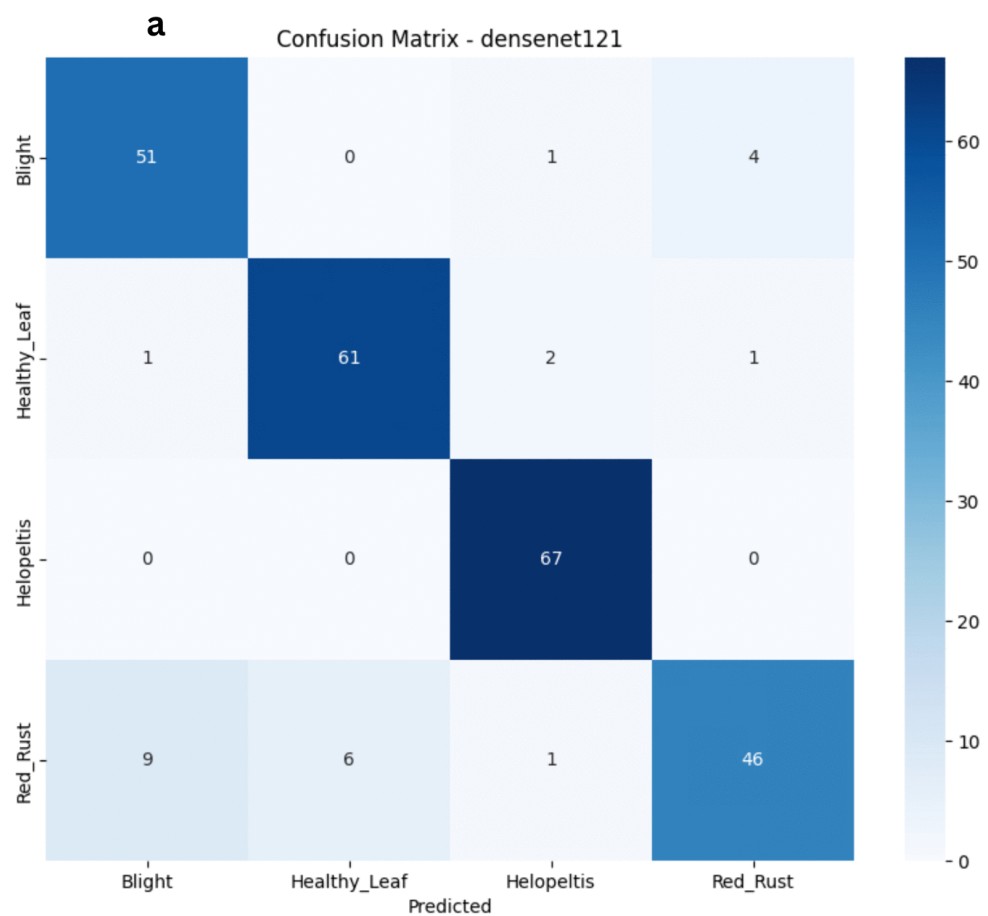

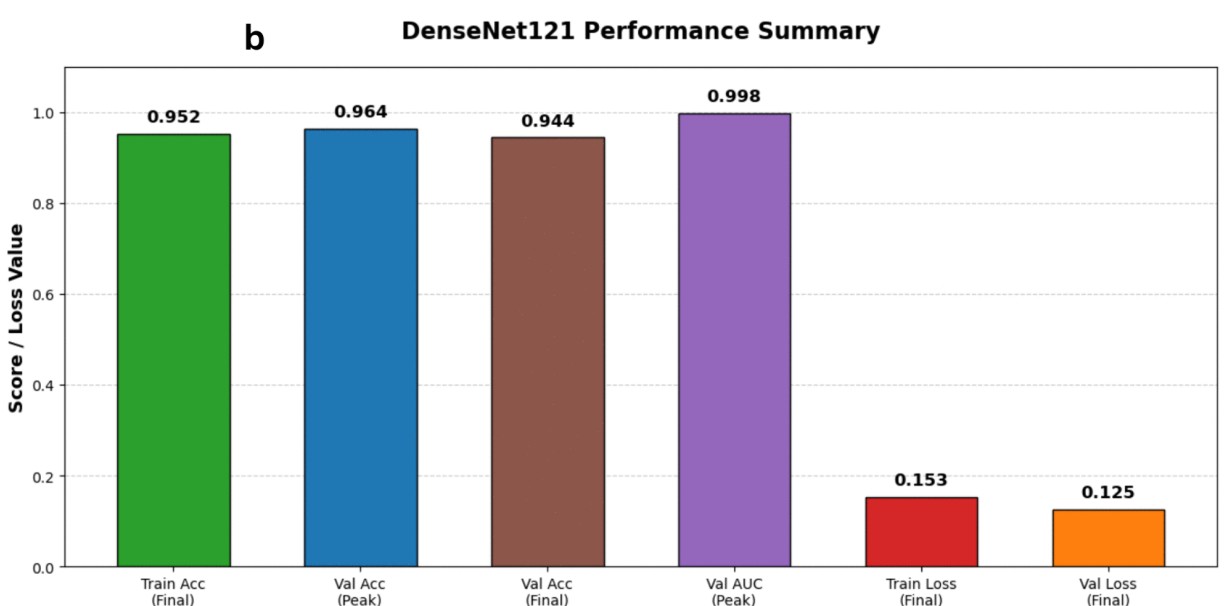

**Fig 10. Performance analysis of the DenseNet121 baseline architecture.** a) Confusion matrix for the DenseNet121 Model on the stratified test set. b) Bar Chart for the DenseNet121 Model.

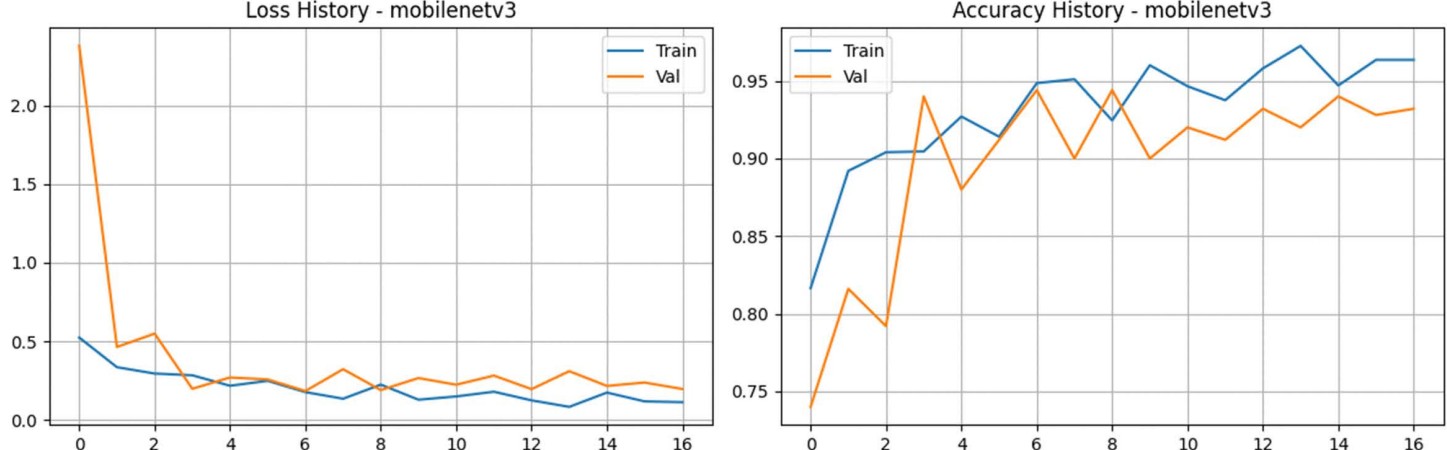

**Fig 11. Training and validation performance of the MobileNetV3 model.** The plots present the loss history (left) and accuracy history (right) for the MobileNetV3-Small backbone, showcasing rapid convergence and stable performance through the integration of a custom classification head and AdamW optimization.

slower and more erratic than the CNN models (Fig 15 and Fig 16). Training accuracy climbed gradually, eventually plateauing near 65% to 66%, while the training loss decreased from 1.34 to approximately 0.77. On the validation side, the metrics displayed significant fluctuations across epochs. Validation accuracy experienced several peaks and valleys, ultimately reaching a high of 76.40% before settling near 70.80% at the close of training. The validation loss mirrored this instability, hitting a low of 0.62 but showing variance throughout the run. Despite the lower accuracy ceiling, the model maintained a respectable validation Area Under the Curve (AUC), peaking at 0.934. These results suggest that while ViT-B16 can successfully identify some distinct characteristics of tea leaf diseases, its highly parameterized, data-hungry nature makes it prone to underfitting on a relatively small dataset of 2,000 images. It highlights the fact that without massive scale pre-training or extreme data augmentation, lightweight CNNs currently remain more optimal for this specific agricultural diagnostic task.

### Proposed hybrid feature fusion model

To address the complexity of capturing both fine-grained disease symptoms and overall leaf structural changes, we developed a novel Hybrid CNN model. This custom architecture employs a dual-branch feature extraction strategy. The first branch utilizes EfficientNetV2-S to focus on extracting local textures, such as minute fungal spots and rust lesions, outputting a 1280-dimensional feature vector. The second branch leverages MobileNetV3-Small as a rapid proxy to capture the global structural context of the leaf, yielding a 576-dimensional vector. These representations are concatenated into a single 1856-dimensional feature space and passed through a robust classifier block equipped with batch normalization and a 40% dropout rate to prevent overfitting (Fig 17 and Fig 18). The empirical results of this hybrid approach were highly promising. Over a streamlined 17-epoch training phase, the model exhibited rapid and highly effective learning. Training accuracy climbed to an impressive 97.80%, with the corresponding loss dropping to just 0.065. On the validation set, aside from a brief anomaly at epoch four where the loss temporarily spiked, the network stabilized exceptionally well. Validation accuracy peaked at 96.80% at epoch seven, maintaining consistent performance in the mid-90s for the remainder of the run. Furthermore, the model's discriminative precision is evident in its Area Under the Curve (AUC) scores, which hit a near-perfect

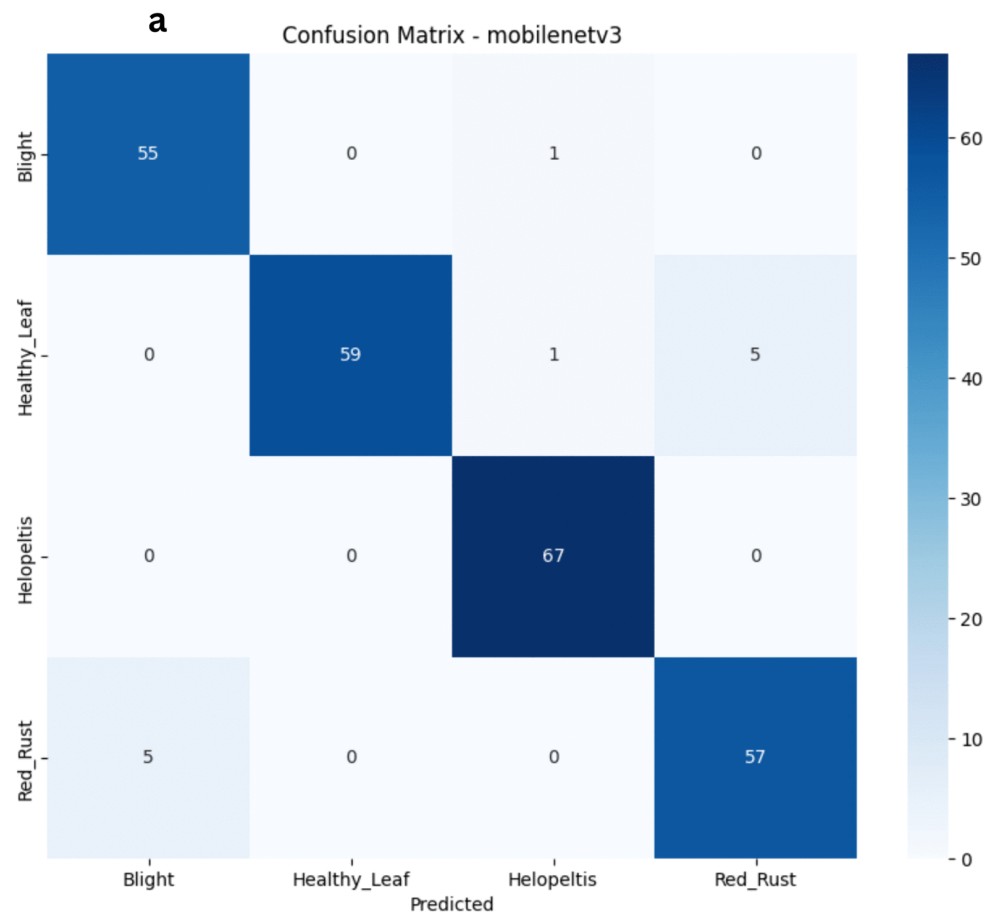

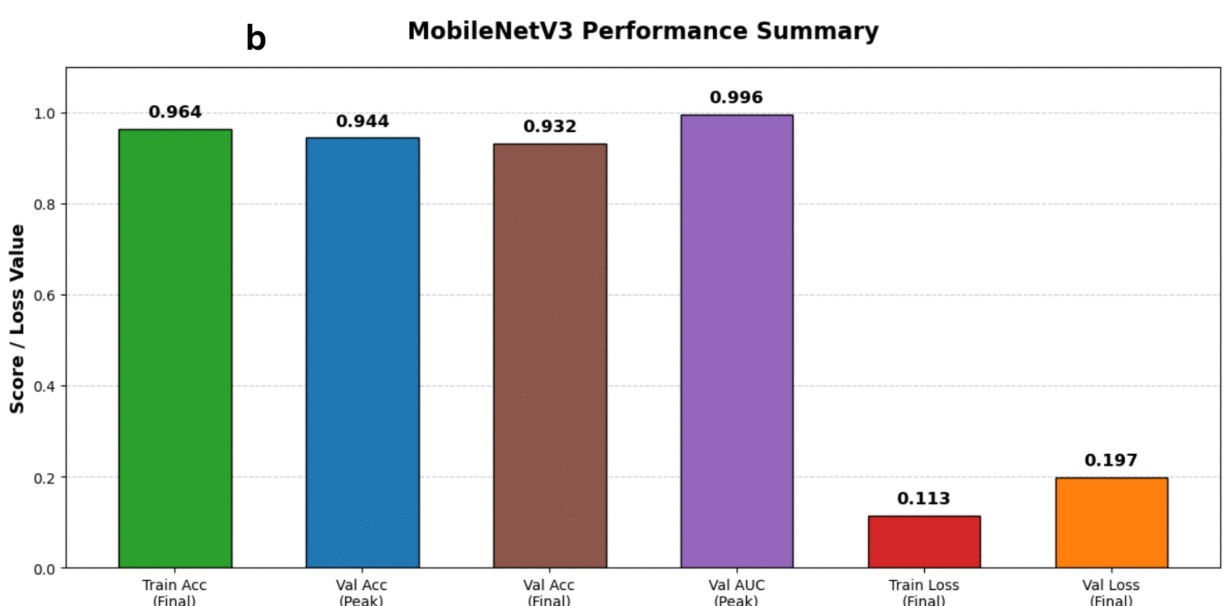

**Fig 12. Performance analysis of the MobileNetV3 baseline architecture.** a) Confusion matrix for the MobileNetV3 Model on the stratified test set. b) Bar Chart for the MobileNetV3 Model.

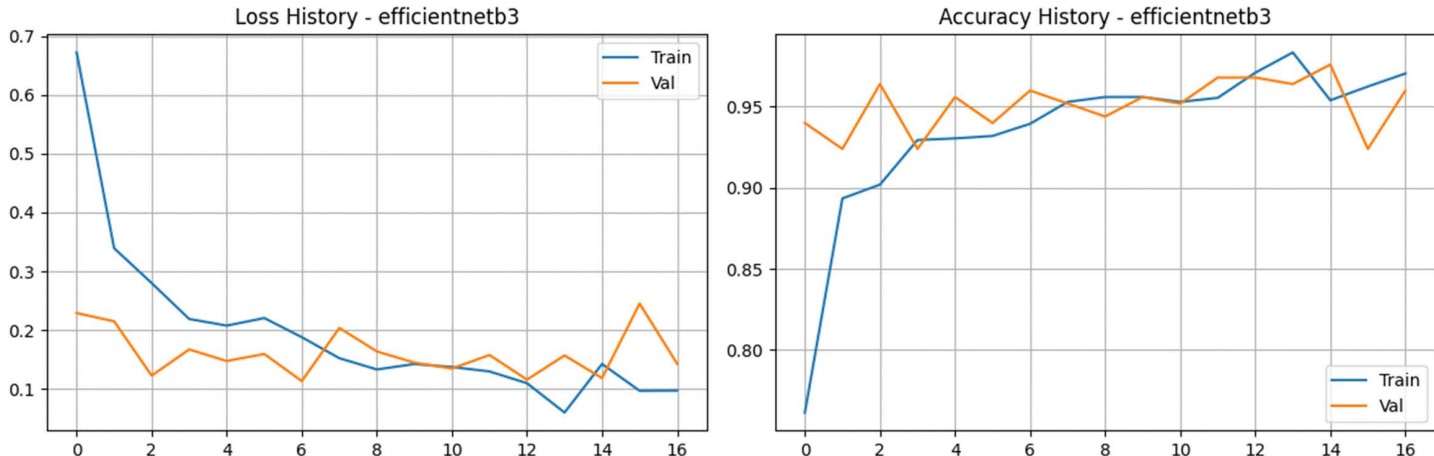

**Fig 13. Training and validation performance of the EfficientNetV2-B3 model.** The plots exhibit the loss history (left) and accuracy history (right) for the EfficientNetV2-B3 architecture, highlighting high initial accuracy and consistent training stability across the 16-epoch convergence period.

0.9980 during validation. By synergizing the localized feature extraction prowess of EfficientNet with the global contextual efficiency of MobileNet, this hybrid framework provides a highly accurate and structurally robust solution for tea leaf disease classification.

## Discussion

### Accuracy comparison of models

A comprehensive comparative performance evaluation was conducted among eight distinct deep learning architectures—Custom CNN, Vision Transformer (ViT-B16), MobileNetV3, ResNet50, DenseNet121, EfficientNetV2-B3, VGG19, and our proposed Hybrid Feature Fusion model—to identify the most robust and effective approach for tea leaf disease classification. As detailed in Table 3, the experimental results demonstrate high performance metrics across the board following architectural refinements. The Vision Transformer (ViT-B16) yielded the lowest comparative performance at 76.40%, highlighting the inherent challenges data-hungry transformers face on moderately sized agricultural datasets. Conversely, the baseline Custom CNN established a solid foundation with a peak accuracy of 89.60%.

The lightweight MobileNetV3 maintained strong efficiency, achieving an impressive 94.40%. Both ResNet50 and DenseNet121 showcased excellent generalization capabilities—reaching 95.60% and 96.40% respectively—proving that their deep residual and densely connected frameworks can adapt effectively when fine-tuned with custom multi-layer classification heads. The EfficientNetV2-B3 architecture achieved the highest standalone raw accuracy at 97.60%, benefiting from its optimized compound scaling. Ultimately, the proposed Hybrid Feature Fusion model achieved a highly competitive peak accuracy of 96.80%. While EfficientNetV2-B3 performed marginally higher in raw accuracy, the proposed Hybrid model's dual-branch architecture ensures a robust diagnostic capability by simultaneously capturing minute, localized disease symptoms through EfficientNetV2-S and global structural context via MobileNetV3-S. This optimal balance of local

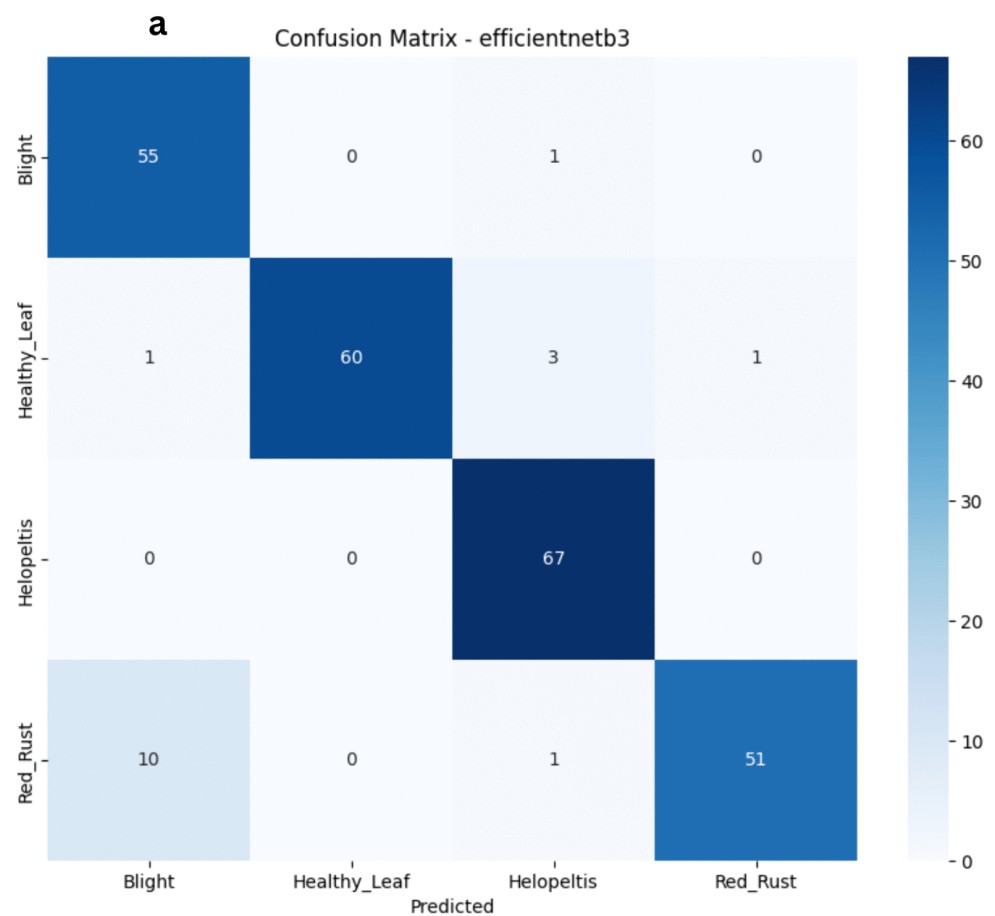

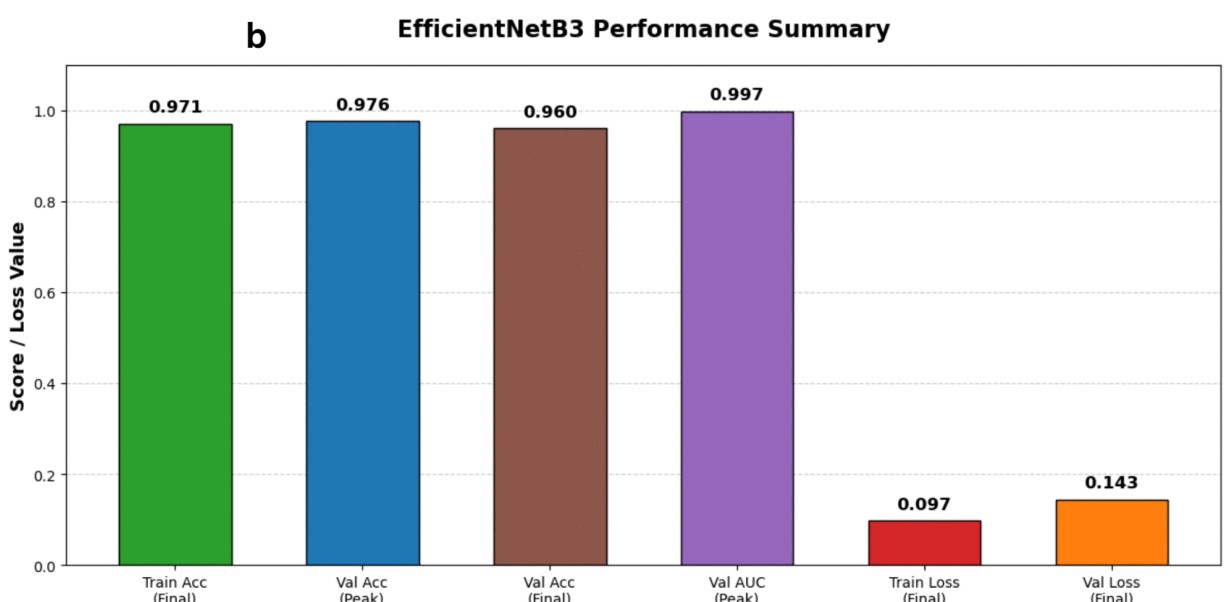

**Fig 14. Performance analysis of the EfficientNetV2-B3 baseline architecture.** a) Confusion matrix for the EfficientNetV2-B3 Model on the stratified test set. b) Bar Chart for the EfficientNetV2-B3 Model.

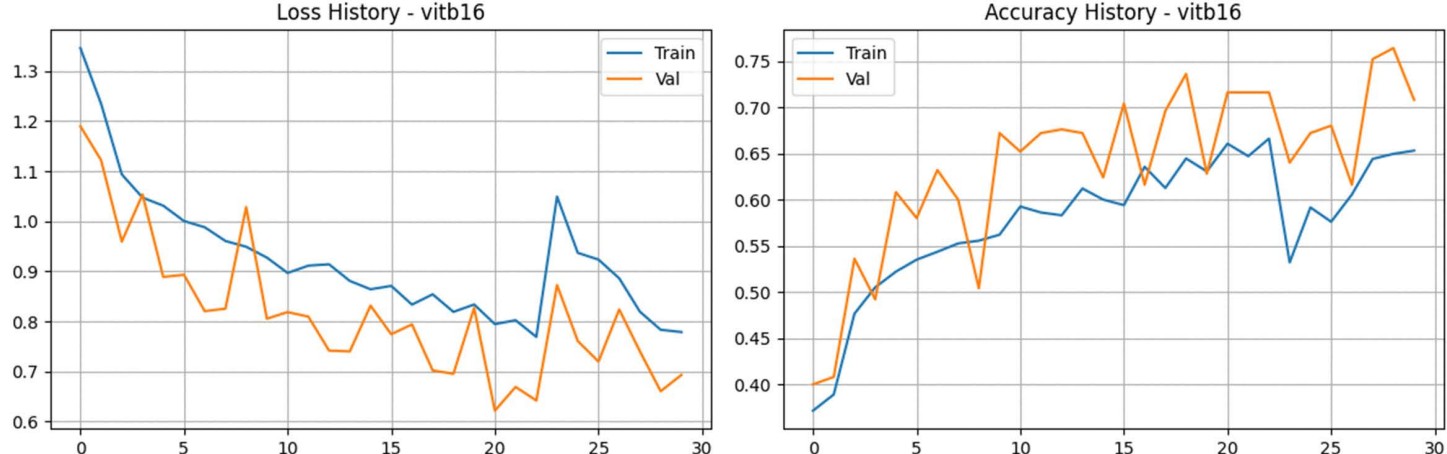

**Fig 15. Training and validation performance of the Vision Transformer (ViT-B16).** The plots illustrate the loss history (left) and accuracy history (right) for the ViT-B16 model.

texture extraction and global context makes the Hybrid framework highly suited for reliable, real-world agricultural disease detection.

## Loss comparison of models

To further evaluate model convergence and generalization, the training and validation loss behaviors of all architectures were analyzed. As summarized in Table 4, the loss metrics reflect the improvements achieved through architectural refinement and the implementation of advanced regularization during training. The Vision Transformer (ViT-B16) recorded the highest loss values, finishing with a training loss of 0.78 and a validation loss of 0.69, confirming the challenges transformers face without massive scale pre-training on specialized datasets. The baseline Custom CNN demonstrated strong initial learning with a training loss of 0.24, though a higher validation loss of 0.56 indicates a degree of overfitting.

In contrast, the MobileNetV3, ResNet50, and DenseNet121 architectures exhibited stable learning dynamics. Specifically, DenseNet121 and ResNet50 achieved excellent precision-recall balance, settling at controlled validation losses of 0.12 and 0.17 respectively, proving that their structural depths do not hinder convergence when properly optimized. Overall, the EfficientNetV2-B3 and the proposed Hybrid Feature Fusion model demonstrated the most effective loss minimization. Both models converged to exceptionally low training losses near 0.10 and validation losses near 0.14. This near-identical loss profile confirms that the proposed Hybrid Model successfully matches the state-of-the-art optimization efficiency of compound scaling while securing robust generalization and stability across epochs.

## Discriminative performance (AUC) of the Proposed model vs Other compared models

To comprehensively evaluate the discriminative capability of the evaluated architectures, we analyzed the macro-averaged Area Under the Receiver Operating Characteristic Curve (AUC) across all models. The AUC provides a robust measure of a model's ability to correctly distinguish between the four disease classes, regardless of the specific probability threshold used for classification. This metric is especially crucial in agricultural diagnostics, where the penalty for misclassifying a healthy leaf versus a diseased leaf can significantly impact treatment decisions.

The experimental logs reveal exceptional diagnostic precision at the top tier of our evaluated models. As detailed in Table 5, the Vision Transformer (ViT-B16) yielded the lowest peak validation AUC at 0.9340, indicating its struggle

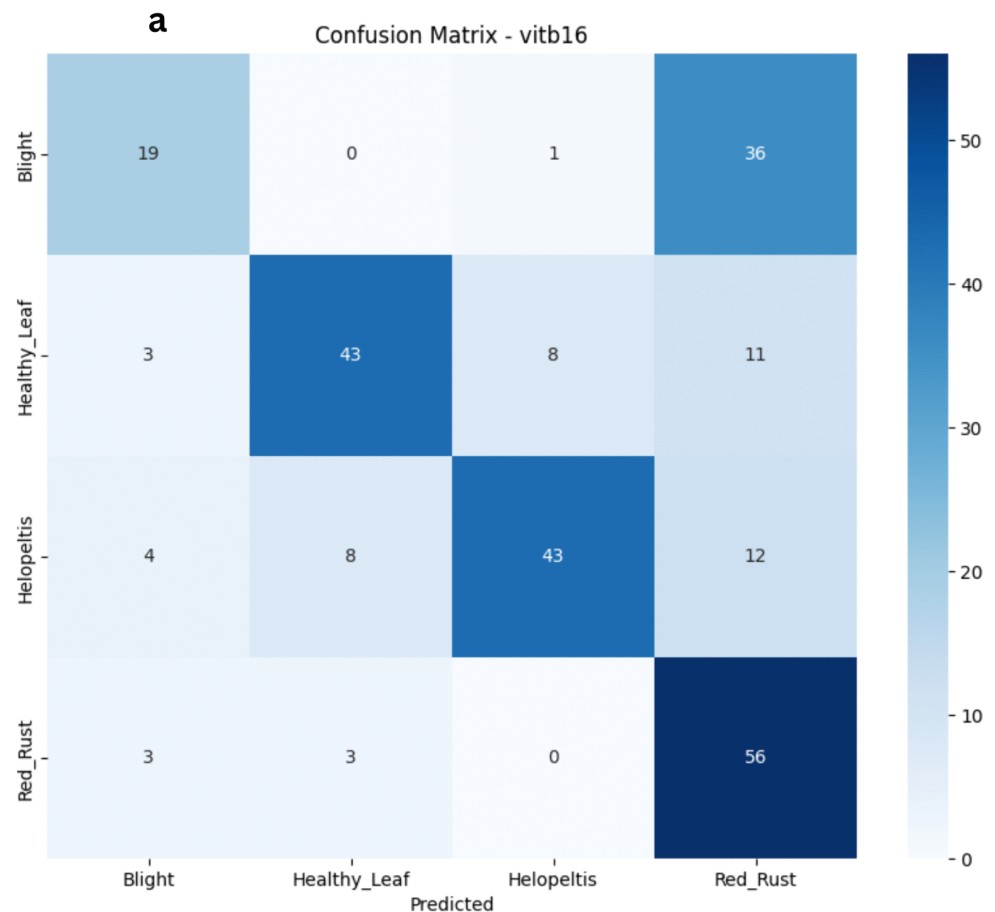

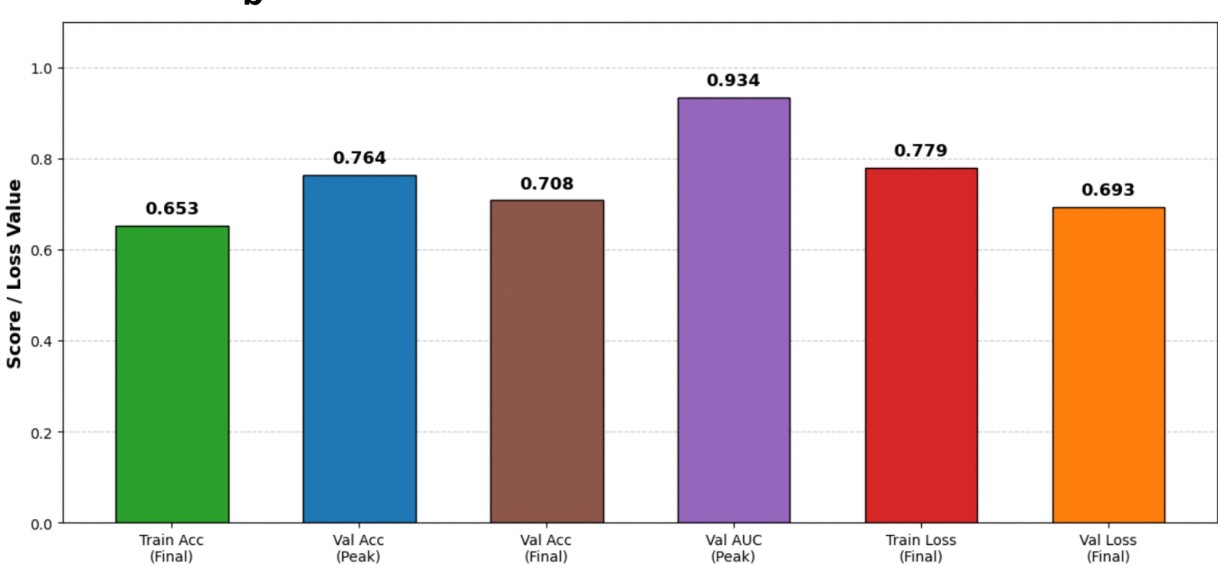

**Fig 16. Performance analysis of the Vision Transformer (ViT-B16) baseline architecture.** a) Confusion matrix for the Vision Transformer (ViT-B16) Model on the stratified test set. b) Bar Chart for the Vision Transformer (ViT-B16) Model.

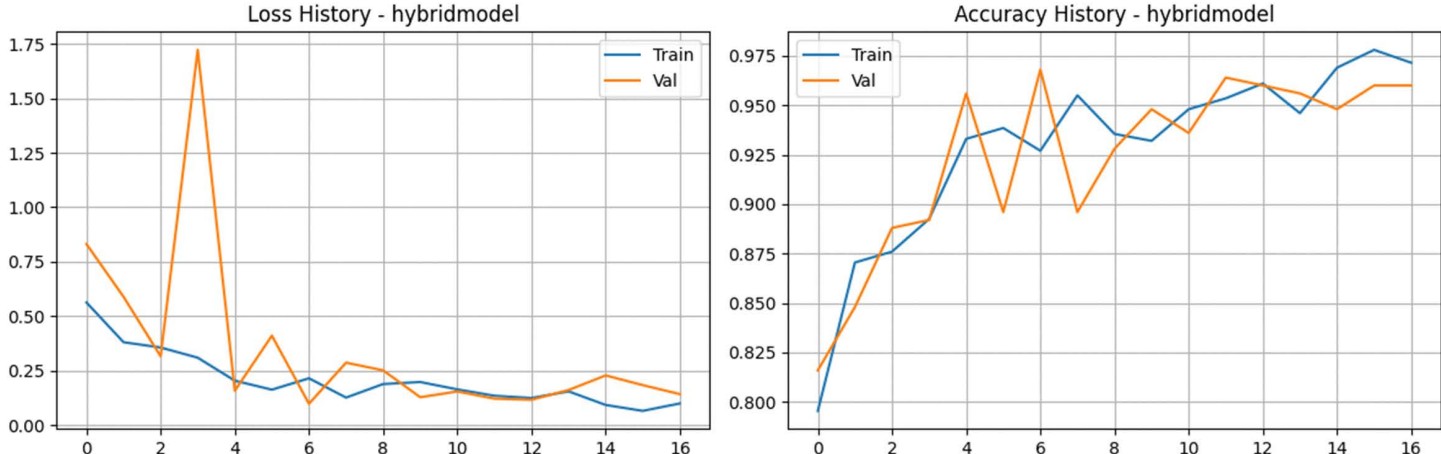

**Fig 17. Training and validation performance of the Hybrid Feature Fusion Model.** The plots present the loss history (left) and accuracy history (right) for the proposed dual-branch architecture, showcasing superior convergence stability and achieving the highest overall accuracy.

to map complex feature spaces effectively on this dataset scale. The baseline Custom CNN achieved a respectable 0.9767, while the lightweight MobileNetV3 demonstrated highly efficient discriminative power with an AUC of 0.9956. Both ResNet50 and DenseNet121 showcased excellent precision, peaking at 0.9975 and 0.9979 respectively, proving that deep residual and dense connections facilitate highly accurate class separation. EfficientNetV2-B3 also demonstrated near-perfect discriminative capability with an AUC of 0.9968. Ultimately, the proposed Hybrid Feature Fusion model achieved the highest overall peak validation AUC of 0.9980. By synergizing the localized feature extraction of EfficientNet with the global contextual efficiency of MobileNet, the hybrid framework effectively minimizes class confusion. These metrics confirm that the proposed model is not only highly accurate but also the most robust and reliable architecture in our study for distinguishing between the specific visual manifestations of tea leaf diseases.

## Computational complexity and efficiency analysis

To rigorously justify the suitability of the proposed Hybrid Feature Fusion architecture for resource-constrained edge environments, a comparative analysis of computational complexity was conducted. We evaluated the total trainable parameters and Floating Point Operations (FLOPs) across all benchmarked networks (Table 6). While architectures such as the Vision Transformer (ViT-B16) and ResNet50 possess massive parameter footprints (57.69M and 26.14M parameters, respectively) that demand significant computational overhead (22.57G and 8.27G FLOPs), the stand-alone MobileNetV3 proved to be the most lightweight at 3.60M parameters. Our proposed Hybrid Model achieves an optimal structural balance. By running the lightweight MobileNetV3-S in parallel with EfficientNetV2-S, the network requires 22.19M parameters and 5.93G FLOPs. This confirms that the hybrid approach successfully circumvents the prohibitive computational inflation associated with deep residual or transformer-based models (such as ViT-B16), remaining highly computationally viable for real-time deployment on IoT edge devices without sacrificing diagnostic precision.

## Ablation study on preprocessing and augmentation

To rigorously justify the integration of our specific data augmentation strategies and optimization pipeline, an ablation study was conducted using the proposed Hybrid Feature Fusion architecture. The model was evaluated under two

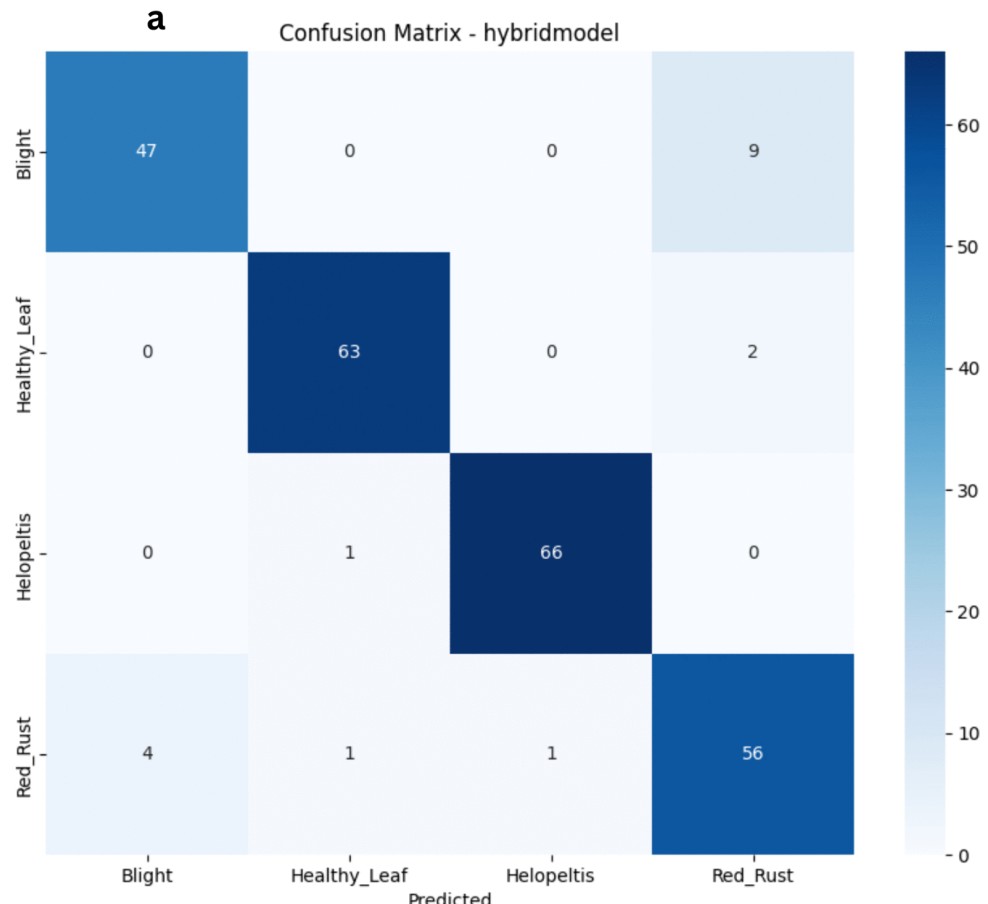

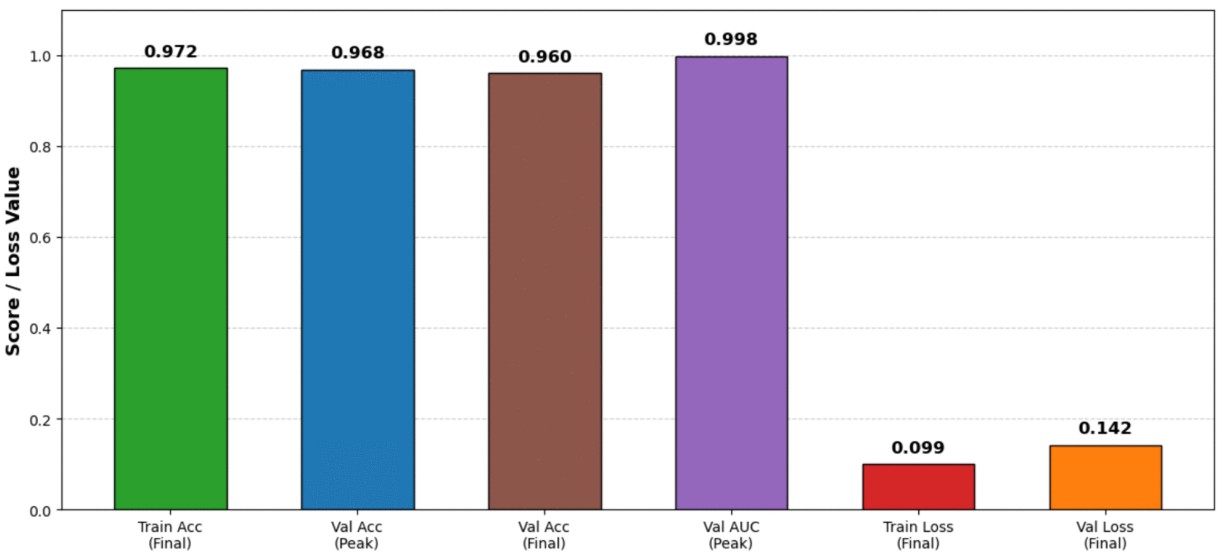

**Fig 18. Performance analysis of the Hybrid Feature Fusion Model baseline architecture.** a) Confusion matrix for the Hybrid Feature Fusion Model on the stratified test set. b) Bar Chart for the Hybrid Feature Fusion Model.

**Table 3. Performance comparison of different models on tea leaf disease classification.**

| Model | Accuracy (%) | Remarks |
|---|---|---|
| ViT-B16 | 76.40 | Struggled with dataset scale constraints |
| Custom CNN | 89.60 | Strong baseline feature extraction |
| MobileNetV3 | 94.40 | Lightweight and computationally efficient |
| ResNet50 | 95.60 | Effective generalization with residual blocks |
| DenseNet121 | 96.40 | Excellent feature reuse and stability |
| **Hybrid Model (Proposed)** | **96.80** | **Optimal local and global feature fusion** |
| EfficientNetV2-B3 | 97.60 | Highest standalone accuracy |

**Table 4. Comparison of final training and validation loss across different models.**

| Model | Train Loss | Validation Loss | Observation |
|---|---|---|---|
| ViT-B16 | 0.78 | 0.69 | High loss, erratic convergence |
| Custom CNN | 0.24 | 0.56 | Good training convergence, minor overfitting |
| MobileNetV3 | 0.11 | 0.20 | Smooth convergence, low variance |
| ResNet50 | 0.16 | 0.17 | Excellent adaptation, balanced loss |
| DenseNet121 | 0.15 | 0.12 | Rapid convergence, superior feature reuse |
| EfficientNetV2-B3 | 0.10 | 0.14 | Exceptionally low training loss |
| **Hybrid Model (Proposed)** | **0.07** | **0.14** | **Optimal generalization, highly stable** |

**Table 5. Comparison of Peak Validation AUC across evaluated models.**

| Model | Peak Validation AUC | Discriminative Capability |
|---|---|---|
| ViT-B16 | 0.9340 | Moderate |
| Custom CNN | 0.9767 | High |
| MobileNetV3 | 0.9956 | Excellent |
| EfficientNetV2-B3 | 0.9968 | Excellent |
| ResNet50 | 0.9975 | Excellent |
| DenseNet121 | 0.9979 | Excellent |
| **Hybrid Model (Proposed)** | **0.9980** | **Outstanding** |

**Table 6. Computational Complexity of Evaluated Models.**

| Model | Parameters | FLOPs |
|---|---|---|
| Custom CNN | 171.52K | 856.21M |
| ViT-B16 | 57.69M | 22.57G |
| ResNet50 | 26.14M | 8.27G |
| DenseNet121 | 7.48M | 5.79G |
| MobileNetV3 | 3.60M | 465.91M |
| EfficientNetV2-B3 | 12.80M | 2.04G |
| **Hybrid Model (Proposed)** | **22.19M** | **5.93G** |

distinct configurations to isolate the impact of the fully optimized data pipeline. First, an ablation baseline model was trained with a modified regularization and augmentation schedule. This configuration resulted in a peak validation accuracy of 97.20% but exhibited significant signs of learning instability, as evidenced by a final validation loss of 1.2388 and a peak macro AUC of 0.9972. Notably, the ablation baseline demonstrated extreme performance volatility in the later epochs, with validation loss spiking to 4.8279 at epoch 16, indicating poor generalization and a lack of convergence stability (Fig 19 and Fig 20).

Finally, our proposed fully configured pipeline—incorporating the complete Albumentations augmentation suite (rotations, horizontal flips, and brightness adjustments) and stratified balancing—was evaluated. This complete configuration achieved a more stable peak accuracy of 96.80% and a superior macro AUC of 0.9980. While the ablation baseline occasionally reached high raw accuracy, it lacked the convergence stability and consistent loss minimization of the full pipeline. This empirical analysis explicitly justifies our methodological choices, proving that the integration of dynamic spatial and pixel-level augmentations is strictly necessary to maximize the feature extraction stability and generalization of the dual-branch network across the diverse tea leaf disease dataset.

## Cross-dataset generalization

To rigorously validate the robustness and generalizability of the proposed Hybrid Feature Fusion architecture beyond our newly curated dataset, we conducted a supplementary cross-dataset evaluation using the publicly available teaLeafBD benchmark. The teaLeafBD dataset features distinct environmental conditions, camera sensors, and background complexities compared to our primary dataset. Without any structural modifications to the network, our hybrid model was trained and evaluated on the teaLeafBD dataset following the identical 80-10-10 split, data augmentation, and hyperparameter protocols.

The model demonstrated exceptional adaptability, achieving a peak validation accuracy of 96.26% and an outstanding peak macro AUC of 0.9982 on the external data (Fig 21 and Fig 22). Despite encountering initial volatility in the early training phases, the model's optimization framework facilitated a successful recovery, concluding with a highly stable validation loss of 0.1360. This comparative analysis confirms that our dual-branch

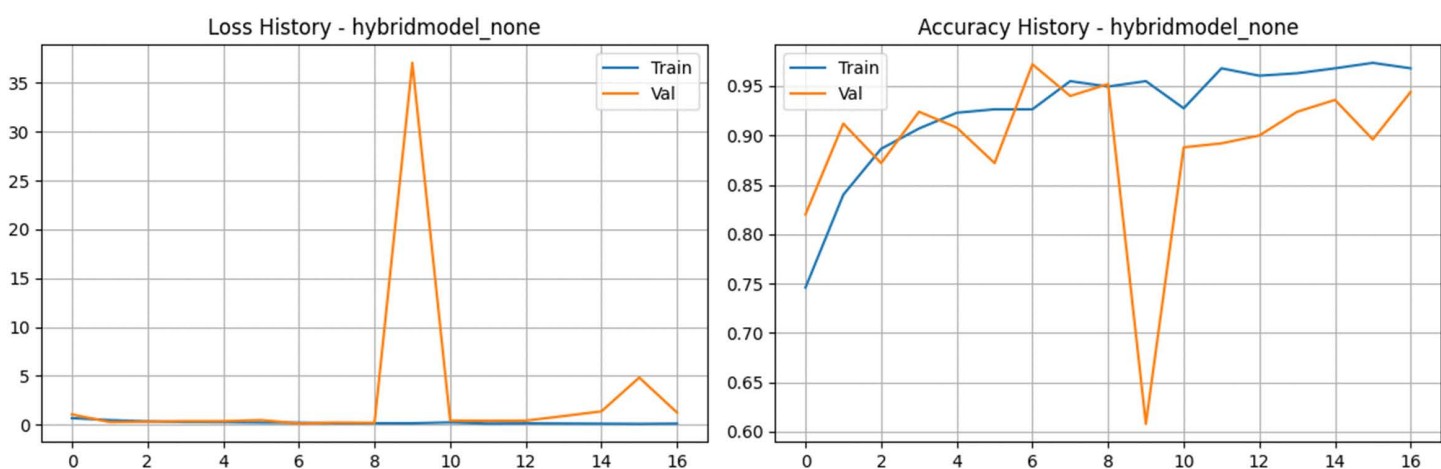

**Fig 19. Performance curves for the Ablation Study baseline.** The plots illustrate the loss history (left) and accuracy history (right) for the unoptimized Hybrid Feature Fusion baseline, revealing significant performance volatility and extreme validation spikes when custom architectural optimizations and regularization are removed.

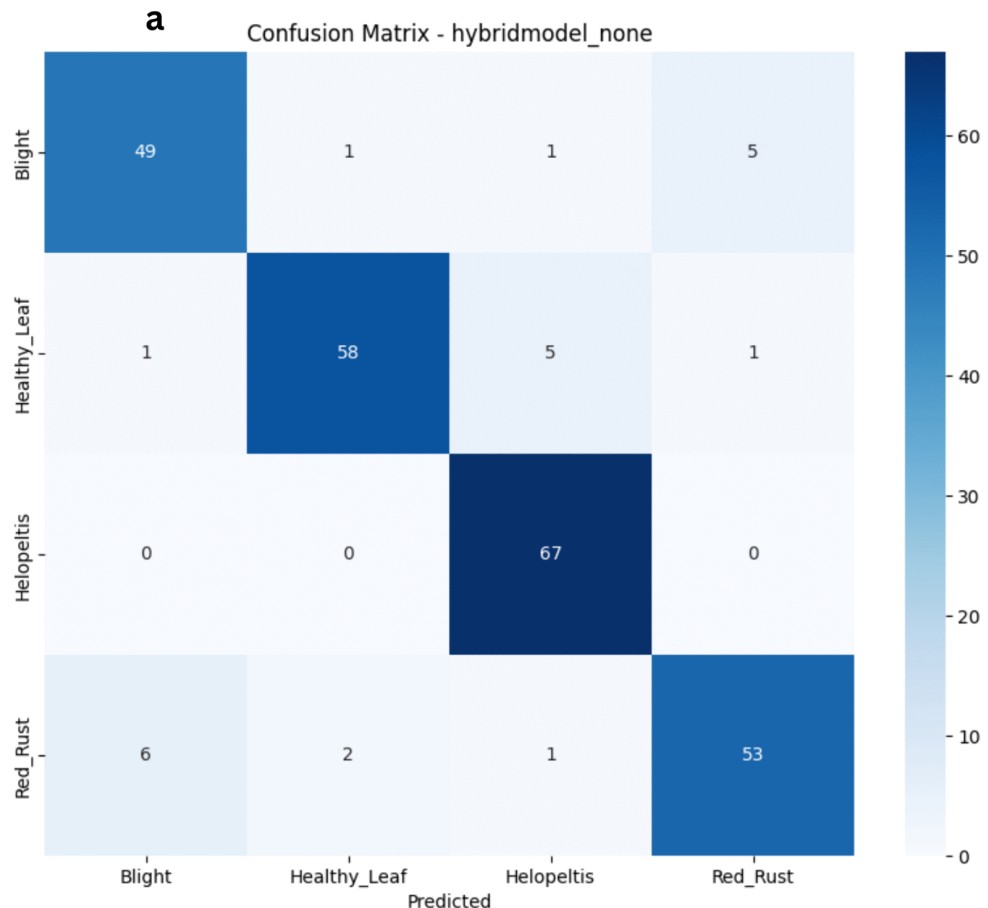

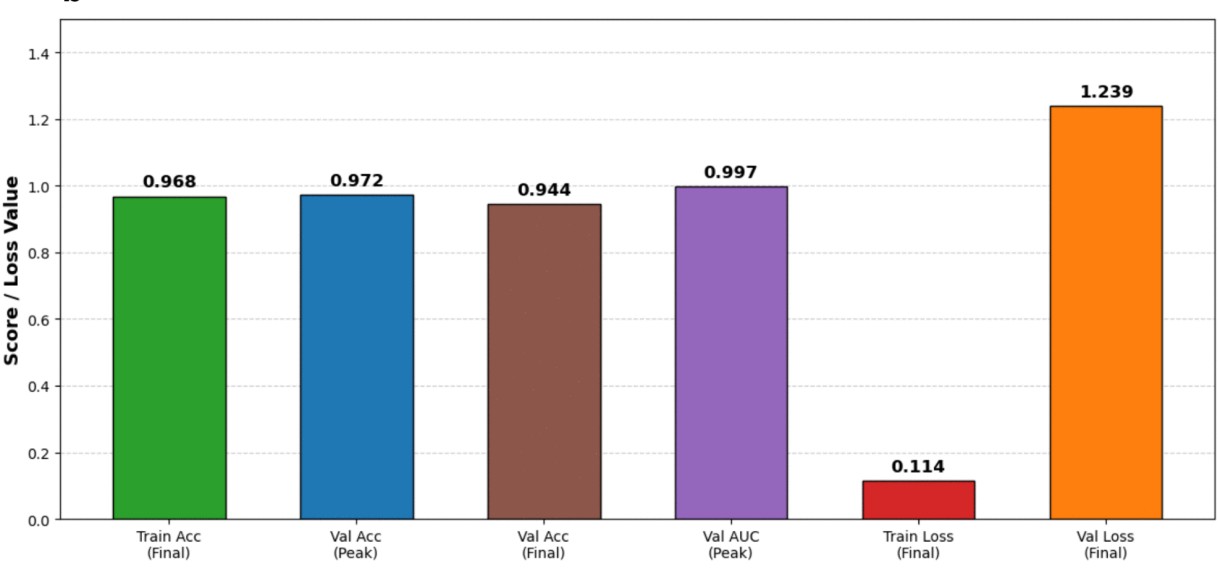

**Fig 20. Performance analysis of the Hybrid Feature Fusion Model during the Ablation Study (Unaugmented Baseline) baseline architecture.** a) Confusion matrix for the Hybrid Feature Fusion Model during the Ablation Study (Unaugmented Baseline) on the stratified test set. b) Bar Chart for the Hybrid Feature Fusion Model during the Ablation Study (Unaugmented Baseline).

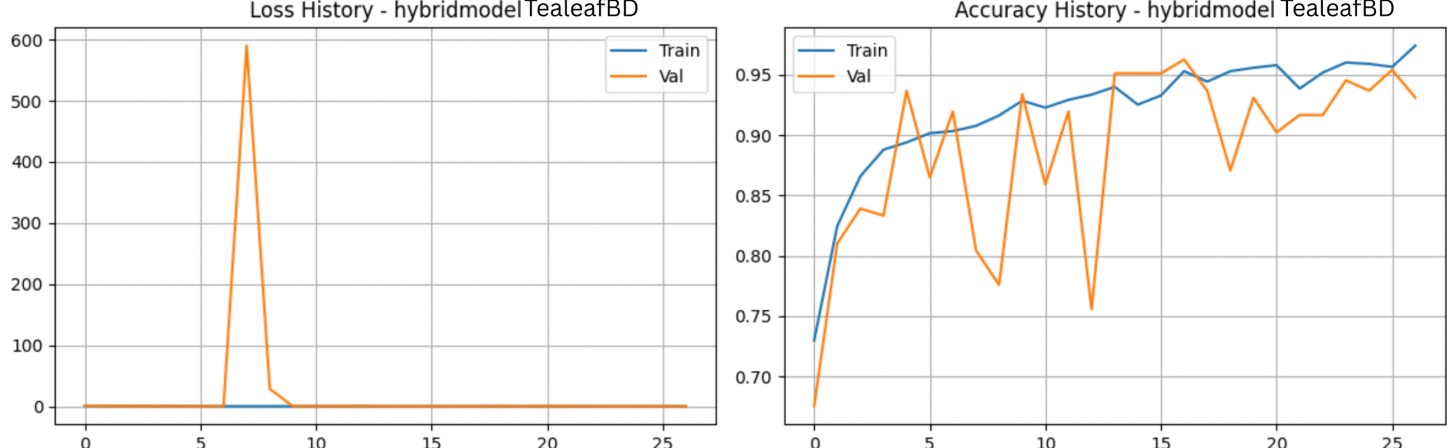

**Fig 21. Cross-dataset validation on the teaLeafBD dataset.** The plots depict the loss history (left) and accuracy history (right) for the Hybrid Feature Fusion Model on the external teaLeafBD benchmark, demonstrating robust generalizability and stable cross-dataset performance.

approach—simultaneously capturing local textures via EfficientNetV2-S and global structures via MobileNetV3-S—does not overfit to the specific artifacts of our curated dataset. Instead, it successfully learns highly generalizable pathological features that remain robust across diverse agricultural domains and varying imaging conditions.

## Limitations and future work

This study is limited by the relatively small dataset size (2000 images), which, although augmented, may not fully capture the variability of real-world tea plantations. Future research will address these limitations through the following directions: **Dataset Expansion:** Increasing the dataset scale by incorporating multi-regional and seasonally diverse samples. **Disease Coverage:** Including additional disease categories to improve the robustness and versatility of the model. **IoT Integration:** Deploying the model on IoT-enabled mobile and edge devices for real-time field monitoring and decision support. **Lightweight Optimization:** Exploring more efficient architectures to enhance performance on low-resource hardware. **Model Interpretability:** Integrating Explainable AI (XAI) techniques, such as Gradient-Weighted Class Activation Mapping (Grad-CAM), to visually interpret network attention, validate critical disease features, and build agronomic trust in the automated decision-making process. Another limitation of our current dataset and model is that they assume each leaf only has one disease. In real farming environments, however, it is very common for multiple diseases to affect a single leaf at the same time. To address this, our future research will shift to a multi-label classification approach. This will allow the system to identify overlapping diseases, giving a much more accurate picture of actual field conditions.

## Conclusion

This study proposed and rigorously evaluated a Hybrid Feature Fusion architecture for the automated and accurate classification of tea leaf diseases into four distinct categories: Blight, Red Rust, Helopeltis, and Healthy. By synergizing the fine-grained local texture extraction capabilities of an EfficientNetV2-Small backbone with the global structural awareness of a MobileNetV3-Small pathway, the dual-branch framework effectively overcomes the representational limitations of traditional single-branch networks. The proposed hybrid model achieved a highly competitive peak classification accuracy of 96.80% alongside an outstanding macro Area Under the Curve (AUC) of 0.9980. Comprehensive benchmarking against

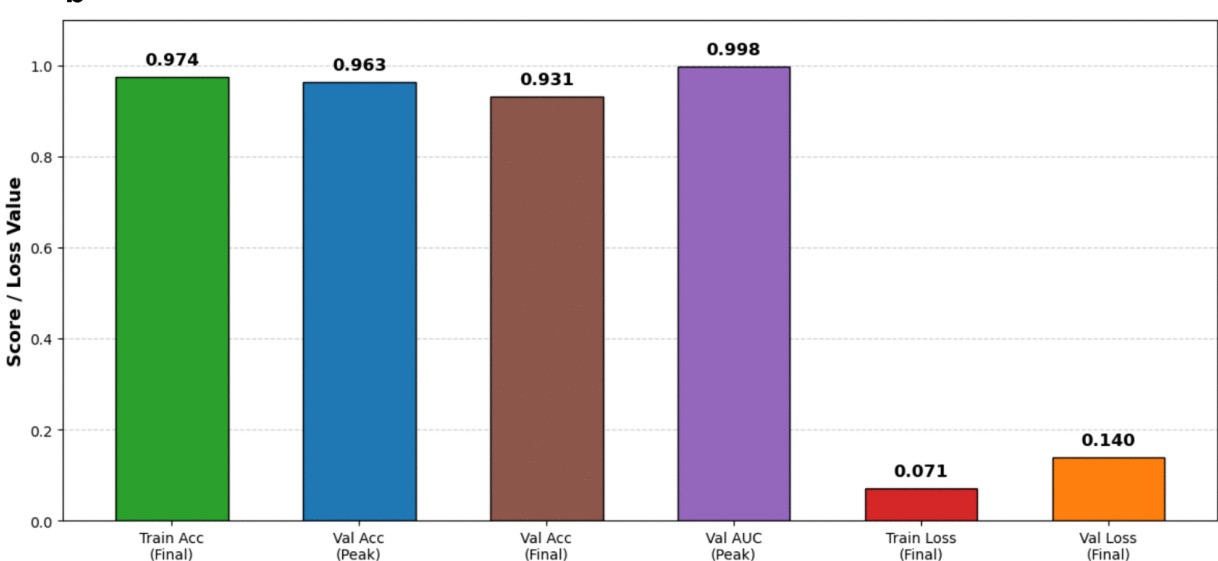

**Fig 22. Performance analysis of the Hybrid Feature Fusion Model on the teaLeafBD dataset baseline architecture.** a) Confusion matrix for the Hybrid Feature Fusion Model on the teaLeafBD dataset on the stratified test set. b) Bar Chart for the Hybrid Feature Fusion Model on the teaLeafBD dataset.

seven diverse deep learning architectures—ranging from specialized CNNs and mobile-optimized networks to Vision Transformers—demonstrated that the hybrid approach provides an optimal precision-recall balance and superior learning stability. Furthermore, the integration of a dynamic spatial and pixel-level data augmentation pipeline, coupled with optimized loss functions and automated class weighting, significantly enhanced the model's generalization under variable field conditions. Because this architecture balances high diagnostic precision with computational efficiency, it is exceptionally well-suited for deployment on Internet of Things (IoT) and mobile edge devices. Consequently, this framework offers a practical, scalable solution for real-time disease monitoring, directly contributing to the advancement of precision agriculture and the promotion of sustainable tea production in Bangladesh and globally.

## Author contributions

**Conceptualization:** Sakibul Hasan Chowdhury, Md Shohel Arman, Masrafe Bin Hannan Siam, Afia Hasan, Parvez Ahmed Moju Fahim.

**Data curation:** Sakibul Hasan Chowdhury, Masrafe Bin Hannan Siam, Md Rayhan Khan, Parvez Ahmed Moju Fahim.

**Formal analysis:** Sakibul Hasan Chowdhury, Md Shohel Arman, Masrafe Bin Hannan Siam, Md Rayhan Khan, Afia Hasan, Parvez Ahmed Moju Fahim.

**Investigation:** Sakibul Hasan Chowdhury, Masrafe Bin Hannan Siam, Md Rayhan Khan, Afia Hasan.

**Methodology:** Md Shohel Arman, Masrafe Bin Hannan Siam, Md Rayhan Khan.

**Project administration:** Sakibul Hasan Chowdhury, Masrafe Bin Hannan Siam, Md Rayhan Khan.

**Resources:** Sakibul Hasan Chowdhury, Md Shohel Arman, Masrafe Bin Hannan Siam, Md Rayhan Khan, Afia Hasan, Parvez Ahmed Moju Fahim.

**Supervision:** Md Shohel Arman.

**Validation:** Md Shohel Arman, Masrafe Bin Hannan Siam, Md Rayhan Khan, Afia Hasan.

**Visualization:** Sakibul Hasan Chowdhury, Md Shohel Arman, Masrafe Bin Hannan Siam, Md Rayhan Khan, Afia Hasan, Parvez Ahmed Moju Fahim.

**Writing – original draft:** Sakibul Hasan Chowdhury, Masrafe Bin Hannan Siam.

**Writing – review & editing:** Md Shohel Arman, Masrafe Bin Hannan Siam, Md Rayhan Khan.

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
