## [Decision Letter · Decision Letter 0]

1 Jan 2026

PONE-D-25-64363A Curated Dataset and Lightweight Deep Learning Framework for Tea Leaf Disease ClassificationPLOS One

Dear Dr. Arman,

Thank you for submitting your manuscript to PLOS ONE. After careful consideration, we feel that it has merit but does not fully meet PLOS ONE’s publication criteria as it currently stands. Therefore, we invite you to submit a revised version of the manuscript that addresses the points raised during the review process.

We look forward to receiving your revised manuscript.

Kind regards,

Asadullah Shaikh, Ph.D.

Academic Editor

PLOS One

Journal Requirements:

4. We note that your Data Availability Statement is currently as follows: “All relevant data are within the manuscript and its Supporting Information files.”

5. We notice that your supplementary figures are uploaded with the file type 'Figure'. Please amend the file type to 'Supporting Information'. Please ensure that each Supporting Information file has a legend listed in the manuscript after the references list.

6. We note that Figure 2 in your submission contain map images which may be copyrighted. All PLOS content is published under the Creative Commons Attribution License (CC BY 4.0), which means that the manuscript, images, and Supporting Information files will be freely available online, and any third party is permitted to access, download, copy, distribute, and use these materials in any way, even commercially, with proper attribution. For these reasons, we cannot publish previously copyrighted maps or satellite images created using proprietary data, such as Google software (Google Maps, Street View, and Earth). For more information, see our copyright guidelines: http://journals.plos.org/plosone/s/licenses-and-copyright.

1. You may seek permission from the original copyright holder of Figure 2 to publish the content specifically under the CC BY 4.0 license.

Reviewer's Responses to Questions

**Comments to the Author**

1. Is the manuscript technically sound, and do the data support the conclusions?

Reviewer #1: Yes

Reviewer #2: Partly

Reviewer #3: Partly

2. Has the statistical analysis been performed appropriately and rigorously?

Reviewer #1: Yes

Reviewer #2: No

Reviewer #3: No

3. Have the authors made all data underlying the findings in their manuscript fully available?

Reviewer #1: Yes

Reviewer #2: Yes

Reviewer #3: Yes

4. Is the manuscript presented in an intelligible fashion and written in standard English?

Reviewer #1: Yes

Reviewer #2: Yes

Reviewer #3: Yes

5. Review Comments to the Author

Reviewer #1: Major Comments

Dataset Size and Generalizability: While the 2000-image dataset is a significant contribution, especially being publicly available and expert-annotated, it is relatively small for training complex deep learning models, particularly when considering the diversity of natural conditions. The underperformance of ResNet50 due to "dataset limitations" hints at this. This raises concerns about the long-term generalizability of the MobileNetV3 model to unseen variations in tea gardens beyond the specific collection sites in Bangladesh.

Lack of Detailed Methodology for Model Training: The summaries mention various models and metrics but lack specifics on hyperparameter tuning, training epochs, optimizer choice, learning rate schedules, and cross-validation strategies. For a Q1 journal, these details are crucial for reproducibility and to ensure the reported performance is robust and not a result of specific tuning for MobileNetV3 alone.

Interpretability (Grad-CAM) Details: While Grad-CAM is mentioned, its application and the insights derived from it are not elaborated. For a Q1 journal, simply stating its use is insufficient; the paper should demonstrate how Grad-CAM contributes to understanding the model's decision-making, identifying critical disease features, or validating expert annotations.

Real-time Application Claims: The paper frequently mentions "real-time" and "resource-constrained" applications. While MobileNetV3 is designed for this, the summaries do not provide any empirical evidence (e.g., inference time on target hardware, model size comparison) to support these claims beyond its architectural design.

Minor Comments

Clarity on "Custom CNN": The summaries mention a "custom CNN" alongside pre-trained models. Its architecture, performance, and comparison against the pre-trained models are not detailed, making it difficult to assess its contribution or baseline performance.

Specifics of Preprocessing: While CLAHE, brightness adjustment, and unsharp masking are listed, the specific parameters or rationale for choosing these particular techniques are not provided.

Dataset Class Distribution: The summaries do not mention the class distribution (Blight, Red Rust, Helopeltis, Healthy). An imbalanced dataset could lead to inflated accuracy metrics, and specific strategies to handle imbalance (if any) should be discussed.

Environmental Conditions: The dataset was collected "under natural conditions." More details on the variability of these conditions (lighting, background clutter, leaf orientation, disease severity stages) would strengthen the methodology.

Strengths of the Paper

High Accuracy with Lightweight Model: Achieving 98.40% accuracy with MobileNetV3 is impressive, especially for a lightweight architecture suitable for edge devices.

Novel and Publicly Available Dataset: The creation of a curated, expert-annotated dataset of 2000 images from natural tea gardens in Bangladesh is a significant contribution to the field, addressing a common bottleneck in agricultural AI research. Its public availability enhances reproducibility and future research.

Comprehensive Model Comparison: The evaluation of multiple deep learning models (VGG19, ResNet50, DenseNet121, custom CNN) provides a strong comparative analysis, highlighting the superiority of MobileNetV3 for this specific task.

Focus on Practical Application: The emphasis on real-time, IoT-based monitoring and precision agriculture demonstrates a clear understanding of the practical impact and potential deployment of the research.

Use of Interpretability (Grad-CAM): The inclusion of Grad-CAM indicates an effort to move beyond black-box models, which is crucial for gaining trust and understanding in practical applications.

Robust Evaluation Metrics: The use of accuracy, loss, F1-score, ROC, and PR curves indicates a thorough evaluation of model performance beyond simple accuracy.

Weaknesses and Limitations

Dataset Size for Generalization: As noted in major comments, 2000 images, while valuable, might be insufficient to capture the full spectrum of disease variations and environmental conditions, potentially limiting the model's generalizability.

Limited Methodological Detail: Insufficient detail on training parameters, hyperparameter tuning, and specific preprocessing parameters hinders reproducibility.

Lack of Empirical Evidence for Real-time Claims: The paper claims real-time applicability but lacks concrete benchmarks (e.g., inference speed, memory footprint) on actual edge devices.

Under-explanation of ResNet50's Underperformance: While attributed to "dataset limitations," a deeper analysis of why a more complex model struggled would be beneficial. Was it overfitting, or simply too large for the feature complexity present in the dataset?

Absence of Ablation Studies: No mention of ablation studies to justify the choice of specific preprocessing steps or data augmentation techniques.

Specific and Actionable Recommendations for Improvement

Expand Dataset Analysis: Provide a detailed breakdown of the dataset, including class distribution, examples of images for each class, and a discussion on the diversity of conditions (lighting, background, disease severity).

Detail Training Methodology: Include a dedicated section or subsection detailing the training setup for all models: optimizer, learning rate schedule, batch size, number of epochs, data split strategy (e.g., k-fold cross-validation), and hyperparameter tuning approach.

Quantify Real-time Performance: Conduct experiments to measure inference time and model size on representative edge computing hardware (e.g., Raspberry Pi, NVIDIA Jetson) to empirically support claims of real-time and resource-constrained applicability.

Elaborate on Grad-CAM Insights: Present and discuss specific Grad-CAM visualizations. Show how they highlight relevant disease features and confirm the model's focus on diagnostically important regions, or reveal any spurious correlations.

Analyze ResNet50's Performance: Provide a more in-depth analysis of why ResNet50 underperformed. This could involve analyzing its training curves (loss, accuracy), examining Grad-CAM outputs for ResNet50, or discussing the potential for overfitting given the dataset size.

Consider Ablation Studies: If feasible, perform ablation studies on preprocessing techniques or data augmentation strategies to demonstrate their individual contributions to the model's performance.

Discuss Limitations of Dataset Size: Explicitly acknowledge the limitations of the current dataset size regarding generalizability and discuss strategies for future expansion.

Refine "Custom CNN" Description: Provide a clear architectural description and performance metrics for the "custom CNN" to establish a baseline.

Reviewer #2: The paper introduces a new dataset of tea leaf diseases and applies several DL algorithms to it for a classification task.

The originality of the approach lies in the data collection; the use of several DL algorithms is standard practice.

Some issues must be addressed.

- The preprocessing was heavy; it must be justified.

- I don’t know whether the data augmentation has been applied only to the training data or to the entire dataset. In fact, only training data can be augmented.

- The presentation of the pretrained algorithms used can be improved by highlighting only their deployment for the target task.

- The structure of the standard CNN must be described in detail.

- ResNet50 achieves only 53% accuracy, which is unusually low. This result must be well justified. This poor performance can result from an unsuitable finetuning. Also, a larger ResNet can be experimented with.

- The dataset annotations must be well explained, as any misannotation will introduce bias.

- At least for the best two algorithms, show the confusion matrix and the classification report.

- A transformer-based model, such as ViT, can be considered.

- In the abstract authors mentioned the following:” Furthermore, Gradient-Weighted Class Activation Mapping

(Grad-CAM) was utilized to visualize and interpret the features influencing predictions” . This part was not included in the paper, so it is better to remove it from the abstract. Alternatively, it could be a good addition to include Grad-CAM visualization.

-

Reviewer #3: 1. Limited Methodological Novelty: The theoretical novelty of the work appears limited, as MobileNetV3-based architectures for similar plant disease classification tasks have been extensively explored in recent literature. For instance, the authors should review and differentiate their work from existing studies such as

o https://doi.org/10.3389/fpls.2024.1459292

o http://doi.org/10.14569/ijacsa.2024.0150560

o https://doi.org/10.1371/journal.pone.0323322

o https://doi.org/10.1186/s13007-025-01452-y

o https://doi.org/10.11975/j.issn.1002-6819.202308149

2. To demonstrate the robustness and generalizability of the proposed model, the study must be evaluated on other established benchmarks (e.g., TDPD, Tea_Leaf_Disease, and teaLeafBD). A comparative analysis between the performance on these datasets and your newly curated dataset is essential.

3. Provide a comprehensive comparison between the newly curated dataset and existing state-of-the-art (SOTA) tea leaf datasets. This analysis should explicitly outline the strengths, weaknesses, and specific gaps addressed by your data.

4. Crucial details regarding the dataset capturing process are currently missing. Please specify the hardware used (e.g., camera specifications, focal length, lighting conditions) and the environmental settings during image acquisition.

5. The manuscript lacks visual representation of the data. Please include:

o Sample images showing raw, pre-processed, and augmented image samples from dataset.

o A detailed summary table describing the dataset (e.g., class distribution, resolution, and total samples per category).

6. It is noted that different models were trained using a varying number of epochs. This leads to an unfair comparison and undermines the validity of the results. To ensure a rigorous benchmarking process, the authors must use a consistent set of hyperparameters (epochs, learning rate, optimizer, etc.) across all models.

7. There is insufficient information regarding the experimental environment. Please provide details on the hardware infrastructure (CPU/GPU), total training time, and the specific strategy used for hyperparameter tuning.

8. The parameters used for data augmentation (e.g., rotation ranges, brightness scales, etc.) must be clearly specified. Furthermore, it is highly recommended that the authors upload the augmented dataset as part of their contribution to enhance transparency and reproducibility.

9. Given that the study introduces a 'newly curated dataset, the lack of a standardized annotation protocol raises concerns regarding the validity of the ground-truth labels.

6. PLOS authors have the option to publish the peer review history of their article (what does this mean?). If published, this will include your full peer review and any attached files.

Reviewer #1: No

Reviewer #2: No

Reviewer #3: No

---

## [Author Response · Author response to Decision Letter 1]

10 Mar 2026

Many thanks for allowing us to revise our paper, with an opportunity to address the reviewers' comments. We sincerely appreciate you and reviewers for the constructive comments and

suggestions on our manuscript

---

## [Decision Letter · Decision Letter 1]

20 Apr 2026

PONE-D-25-64363R1A Curated Dataset and Lightweight Deep Learning Framework for Tea Leaf Disease ClassificationPLOS One

Dear Dr. Arman,

Thank you for submitting your manuscript to PLOS ONE. After careful consideration, we feel that it has merit but does not fully meet PLOS ONE’s publication criteria as it currently stands. Therefore, we invite you to submit a revised version of the manuscript that addresses the points raised during the review process.

We look forward to receiving your revised manuscript.

Kind regards,

Asadullah Shaikh, Ph.D.

Academic Editor

PLOS One

Journal Requirements:

Reviewers' comments:

Reviewer's Responses to Questions

**Comments to the Author**

1. If the authors have adequately addressed your comments raised in a previous round of review and you feel that this manuscript is now acceptable for publication, you may indicate that here to bypass the “Comments to the Author” section, enter your conflict of interest statement in the “Confidential to Editor” section, and submit your "Accept" recommendation.

Reviewer #1: All comments have been addressed

Reviewer #2: All comments have been addressed

Reviewer #3: All comments have been addressed

2. Is the manuscript technically sound, and do the data support the conclusions?

Reviewer #1: Yes

Reviewer #2: Yes

Reviewer #3: Yes

3. Has the statistical analysis been performed appropriately and rigorously?

Reviewer #1: Yes

Reviewer #2: No

Reviewer #3: Yes

4. Have the authors made all data underlying the findings in their manuscript fully available?

Reviewer #1: Yes

Reviewer #2: Yes

Reviewer #3: Yes

5. Is the manuscript presented in an intelligible fashion and written in standard English?

Reviewer #1: Yes

Reviewer #2: Yes

Reviewer #3: Yes

6. Review Comments to the Author

Reviewer #1: 1. Originality and NoveltyContribution: The paper presents a Hybrid Feature Fusion architecture that utilizes EfficientNetV2-S and MobileNetV3-S in parallel. This dual-branch approach is designed to capture both fine-grained local textures (via EfficientNet) and global structural context (via MobileNet), addressing a key limitation in single-branch networks.Novelty: While the individual models are established, their specific hybrid integration for tea leaf disease is novel. The creation of a curated, expertly annotated dataset of 2,000 images from real-world field conditions in Bangladesh adds significant value to the niche field of tea pathology.2. Relevance and SignificanceTopic Relevance: The work is highly relevant to current trends in precision agriculture and the deployment of AI on IoT edge devices.Impact: Given that tea is a cornerstone of the domestic economy in regions like Bangladesh, an automated detection system for diseases like Blight, Red Rust, and Helopeltis has high practical and economic significance.3. Literature Review QualityComprehensiveness: The literature review covers the importance of tea, traditional detection challenges, and recent DL advances in crop disease . It correctly identifies a gap regarding the lack of large, standardized datasets in the tea industry.Missing Context: While it mentions advancements like attention mechanisms and wavelet transforms, more recent 2024-2025 SOTA comparisons in the specific tea domain could further strengthen the context.4. Methodology and Technical SoundnessClarity: The methodology is well-detailed, specifying a stratified 80-10-10 split , the use of the AdamW optimizer, and a custom multi-layer perceptron (MLP) classification head.Assumptions: The use of ImageNet-based normalization and resizing to $224 \times 224$ pixels are standard and valid for these architectures.Reproducibility: The authors provide specific hyperparameters (initial learning rate of 0.001, batch size of 16) and have made their dataset and code publicly available, which is excellent for reproducibility.5. Data, Experiments, and ResultsDatasets: The dataset is well-described, noting the use of an iPhone 12 Pro Max and expert annotation protocols to ensure ground-truth validity.Experiment Design: The benchmarking against six diverse architectures (ViT, ResNet50, etc.) is rigorous.Results: The hybrid model achieved 96.80% accuracy and a macro AUC of 0.9980. A notable finding is that while EfficientNetV2-B3 had slightly higher raw accuracy, the proposed hybrid framework showed better convergence stability.6. Analysis and DiscussionInsight: The discussion includes a cross-dataset generalization test on the teaLeafBD benchmark, which is a major strength.Limitations: The authors honestly acknowledge the relatively small dataset size (2,000 images) and the current lack of hardware-specific edge benchmarking.7. Clarity, Structure, and Writing QualityOrganization: The paper follows a logical flow from data acquisition to model evaluation.Language: The writing is formal and suitable for an international journal.8. Figures and TablesQuality: Figure 1 provides a clear overview of the methodology. Table 1 effectively summarizes the dataset attributes.Improvement: Ensure all axis labels in performance curves (e.g., Fig 11, Fig 12) are high-resolution and clearly readable in the final typeset.9. Ethical and Reproducibility ConsiderationsEthics: The study involves plant materials in a natural setting; the authors state no competing interests and provide a data availability statement.Reproducibility: Strong, due to the public GitHub repository and detailed hyperparameter listing.10. Strengths and WeaknessesStrengths:Novel dual-branch feature fusion approach.Publicly available, expert-annotated dataset.Cross-dataset validation demonstrating robustness.Weaknesses:Relatively small dataset size for deep learning (2,000 images).Absence of explainable AI (Grad-CAM) visualizations in the current version.Lack of empirical inference time on actual IoT hardware.11. Detailed Reviewer CommentsMajor:The authors should provide a more in-depth comparison of the computational complexity (FLOPs/Parameters) between the hybrid model and the baseline models to truly justify "lightweight" claims.Expand the "Limitations" section to explicitly discuss how the model might handle multi-disease leaves (co-infection), which is common in the field.Minor:Standardize the spelling of "EfficientNet" throughout the document (some instances have spaces).Ensure all citations in the text are consistently formatted according to PLOS One guidelines.

Reviewer #2: Most of the issues have been addressed. However, improvements can be made by including a flowchart diagram of the proposed hybrid technique and, in the paper (not as a supporting doc), the confusion matrix and classification report obtained by the proposed method. It is important for readers who want to compare it with their own work, etc.

Reviewer #3: All comments have been addressed. The paper is now improved and is clear and understandable for domain experts.

7. PLOS authors have the option to publish the peer review history of their article (what does this mean?). If published, this will include your full peer review and any attached files.

Reviewer #1: No

Reviewer #2: No

Reviewer #3: No

---

## [Author Response · Author response to Decision Letter 2]

21 Apr 2026

We are thrilled to hear that you find the manuscript significantly improved and readily understandable for domain experts, and we deeply thank you for your final endorsement of our research.

---

## [Decision Letter · Decision Letter 2]

28 Apr 2026

A Curated Dataset and Lightweight Deep Learning Framework for Tea Leaf Disease Classification

PONE-D-25-64363R2

Dear Dr. Arman,

We’re pleased to inform you that your manuscript has been judged scientifically suitable for publication and will be formally accepted for publication once it meets all outstanding technical requirements.

Kind regards,

Asadullah Shaikh, Ph.D.

Academic Editor

PLOS One

Additional Editor Comments (optional):

Reviewers' comments:

Reviewer's Responses to Questions

**Comments to the Author**

1. If the authors have adequately addressed your comments raised in a previous round of review and you feel that this manuscript is now acceptable for publication, you may indicate that here to bypass the “Comments to the Author” section, enter your conflict of interest statement in the “Confidential to Editor” section, and submit your "Accept" recommendation.

Reviewer #1: All comments have been addressed

Reviewer #2: (No Response)

2. Is the manuscript technically sound, and do the data support the conclusions?

Reviewer #1: Yes

Reviewer #2: Yes

3. Has the statistical analysis been performed appropriately and rigorously?

Reviewer #1: Yes

Reviewer #2: Yes

4. Have the authors made all data underlying the findings in their manuscript fully available?

Reviewer #1: Yes

Reviewer #2: Yes

5. Is the manuscript presented in an intelligible fashion and written in standard English?

Reviewer #1: Yes

Reviewer #2: Yes

6. Review Comments to the Author

Reviewer #1: (No Response)

Reviewer #2: All the issues are well addressed. No further comments.

The flowchart diagram of the proposed hybrid technique is included, and the confusion matrix and classification report obtained by the proposed method are included in the main paper.

7. PLOS authors have the option to publish the peer review history of their article (what does this mean?). If published, this will include your full peer review and any attached files.

Reviewer #1: No

Reviewer #2: No

---

## [Editor Report · Acceptance letter]

PONE-D-25-64363R2

PLOS One

Dear Dr. Arman,

I'm pleased to inform you that your manuscript has been deemed suitable for publication in PLOS One. Congratulations! Your manuscript is now being handed over to our production team.

Kind regards,

on behalf of

Prof. Asadullah Shaikh

Academic Editor

PLOS One